# Implicit Reward as the Bridge: A Unified View of SFT and DPO Connections

**Bo Wang**[1][*]    **Qinyuan Cheng**[1][*]    **Runyu Peng**[1][*]    **Rong Bao**[1]    **Peiji Li**[1]
**Qipeng Guo**[2]    **Linyang Li**[2]    **Zhiyuan Zeng**[1]    **Yunhua Zhou**[2]    **Xipeng Qiu**[1][†]

[1]School of Computer Science, Fudan University    [2]Shanghai Artificial Intelligence Laboratory

{bwang22, chengqy21, rypeng22, rbao22}@m.fudan.edu.cn
{linyangli19, xpqiu}@fudan.edu.cn
{guoqipeng, zhouyunhua}@pjlab.org.cn

## Abstract

Post-training processes are essential phases in grounding pre-trained language models to real-world tasks, with learning from demonstrations or preference signals playing a crucial role in this adaptation. We present a unified theoretical framework bridging Supervised Fine-Tuning (SFT) and preference learning in Large Language Model (LLM) post-training. Through rigorous mathematical derivation, we demonstrate that both SFT and preference learning methods like Direct Preference Optimization (DPO) operate within the same optimal policy-reward subspace, with SFT representing a special case of implicit reward learning. Our analysis reveals a critical limitation in conventional SFT: the KL divergence term in distribution matching becomes constant with respect to the policy during optimization, failing to constrain model updates. To address this, we propose a simple yet effective learning rate reduction approach that yields significant performance improvements (up to **25%** relative gain and **6%** absolute win rate increase in instruction following tasks. Additionally, we derive alternative SFT objectives from various f-divergence functions that preserve the KL term during optimization, further enhancing post-DPO model performance. Finally, we extend the theoretical relationship between LLM logits and Q-functions from preference learning to the SFT context, providing mathematical derivations and experimental validation.

## 1 Introduction

Post-training represents a critical phase in grounding Large Language Models (LLMs) in real-world applications. After accumulating general prior knowledge from numerous pre-training corpora, post-training aims to leverage the potential of LLMs for different needs, such as following natural language instructions [1, 2, 3, 4, 5]. Two principal methodological approaches dominate the post-training landscape. The first approach learns from expert demonstrations [6, 7], commonly known as imitation learning, which in the context of LLMs is typically referred to as Supervised Fine-Tuning (SFT). The second approach focuses on learning from environmental signals, primarily through Reinforcement Learning methods [8, 9, 10].

Within the post-training landscape, preference signals have emerged as particularly valuable forms of feedback, attracting substantial research attention[11, 1]. Preference learning typically follows a two-stage process (hereafter referred to as **sequential training**): an initial stage of SFT followed by preference optimization methods like Direct Preference Optimization (DPO) [12]. However, the relationship between these critical stages remains predominantly understood through empirical

---

[*]Equal Contribution.
[†]Corresponding author

39th Conference on Neural Information Processing Systems (NeurIPS 2025).

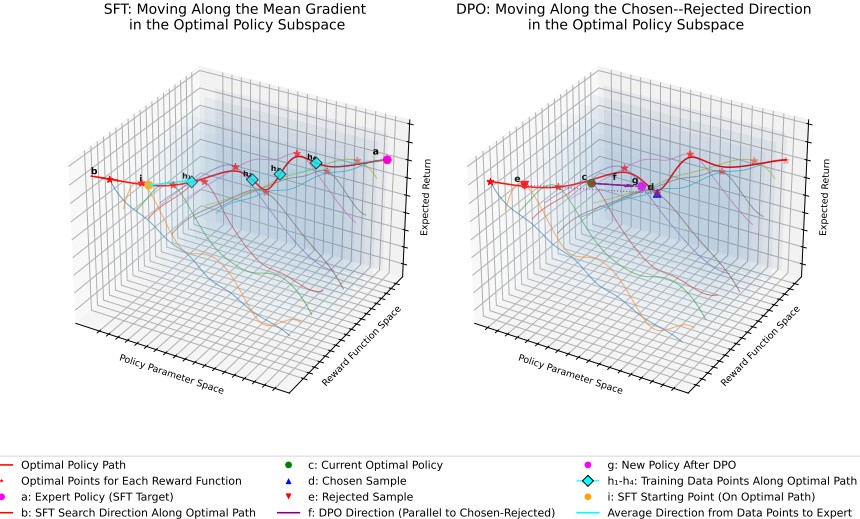

SFT: Moving Along the Mean Gradient in the Optimal Policy Subspace

DPO: Moving Along the Chosen--Rejected Direction in the Optimal Policy Subspace

**Legend:**
- Optimal Policy Path
- Optimal Points for Each Reward Function
- a: Expert Policy (SFT Target)
- b: SFT Search Direction Along Optimal Path
- c: Current Optimal Policy
- d: Chosen Sample
- e: Rejected Sample
- f: DPO Direction (Parallel to Chosen-Rejected)
- g: New Policy After DPO
- $h_1$-$h_4$: Training Data Points Along Optimal Path
- i: SFT Starting Point (On Optimal Path)
- Average Direction from Data Points to Expert

Figure 1: Schematic diagram: SFT and DPO are optimizing implicit rewards in the optimal policy subspace. The $x$-dimension refers to the space of all possible policy models. The $y$-dimension represents the space of all possible reward functions. Given a specific reward function, each possible policy will have its own expected return, forming a curve in the policy-return plane. The optimal policies corresponding to different reward functions constitute a subspace. **Left**: The SFT process searches within this subspace and moves along the average direction indicated by the demonstrations. **Right**: Likewise, DPO operates within this same subspace, but navigates along the direction vector formed by the chosen and rejected examples.

observations rather than theoretical foundations, and SFT is often treated merely as a preparatory warm-up step [13]. Despite the widespread adoption of this sequential paradigm, a significant gap persists in theoretical perspectives regarding how these two approaches fundamentally relate to one another. While previous research [14] has extensively explored various aspects of LLM learning dynamics, the theoretical connections between SFT objectives and preference learning frameworks have received insufficient attention, limiting our understanding of their combined effectiveness in the post-training process.

To mitigate this gap, we prove that implicit reward learning can be utilized as a unified view connecting SFT and preference-learning processes. Previous work [12] established that preference learning in the second stage can operate through implicit rewards. In our research, we revisit the distribution matching objective and apply necessary adjustments for post-training. We provide a comprehensive mathematical proof following earlier works [15, 16]. Our proof demonstrates that the conventional SFT objective represents a special case of learning implicit rewards. Figure 1 illustrates our theoretical conclusion. The optimal policy for each possible reward function forms a policy-reward subspace, with both SFT and DPO operating within this subspace.

This theoretical framework of implicit rewards yields several novel insights. By framing SFT as a training target derived from distribution matching, we uncover a key insight: the crucial KL term in the objective functions merely serves as a zero-order component. Since this term remains constant with respect to $\pi$, it imposes no constraints on model updates following differentiation. We propose a simple yet efficient heuristic to mitigate this issue by reducing the learning rate. Furthermore, we identify alternative training objectives by choosing different $f$-divergence derivative functions for distribution matching that preserve the KL term during optimization and show their effectiveness empirically. Finally, we demonstrate that LLM logits can function as a Q-function corresponding to implicit rewards during the SFT process. This extends the theoretical framework in [17], which primarily established this relationship in the DPO setting, while our work reveals similar mathematical structures within the SFT process. Our empirical training results align strongly with these theoretical predictions in the instruction-following tasks.

Our main contributions are as follows:

1. We revisit the distribution matching objective and mathematically prove that SFT also learns an implicit reward function identical to that of DPO. This provides a unified theoretical view that clarifies the relationship between SFT and preference learning.

2. Within this theoretical framework, we provide a simple yet effective approach by reducing the learning rate during the SFT phase to mitigate the absence of the KL term. The KL term typically ensures that the policy does not deviate excessively from the base model, promoting stable and efficient learning. This significantly improves results, with relative improvements up to 25% (absolute win rate increases of up to 6%, from 15.6% to 21.5%).

3. We also propose several alternative SFT objectives derived from other $f$-divergence functions for LLMs. We demonstrate further improvements in model performance after DPO training (hereafter referred to as **post-DPO**), which yields up to 4% absolute win rate improvements.

4. We mathematically extend the relationship between LLM logits and Q-functions from the DPO context to the SFT process, supporting this extension with indirect experimental evidence. This formulation enables us to efficiently estimate state values under the model's implicit reward and provides deeper insights into the role of SFT in the alignment process.

## 2   Related Work

### 2.1   Inverse Reinforcement Learning and $f$-Divergence

First formalized by [18], Inverse Reinforcement Learning assumes that the expert represents the optimal policy under a certain reward function. It tries to recover this reward function and train a policy model that maximizes it. This approach was significantly advanced by [19] with the principle of maximum entropy to ensure proper exploration. Some commonly used IRL methods include game-theoretic approaches, most notably GAIL [20]. It trains in an adversarial manner using explicit reward recovery and learns policies from the recovered reward function. [21] mathematically provides a unified view using $f$-divergence. [22] also leverages different forms of $f$-divergence derived for imitation learning. It involves iterative optimization and primarily explores applications in classical RL scenarios.

Learning an explicit reward function often involves additional parameters and high-variance adversarial optimization. [15] avoids explicit reward learning by learning a Q-function parameterized by both reward and policy. [16] followed this work and introduced non-adversarial imitation learning for large language models. [23] formulates imitation learning using reverse KL divergence with an entropy regularizer and avoids explicitly training a reward function, which helps reduce overfitting. [24] involves different formulations of $f$-divergence in solving the out-of-distribution (OOD) problem in the generation process and enables backtracking in sequence generation.

Instead of using the Bellman equation like [15, 16], we apply closed-form solutions and avoid inner-loop optimization. In comparison with [23], our work utilizes a more general $f$-divergence formulation [21] and does not introduce negative samples. Unlike [24], we primarily examine the relationship between SFT and DPO in the sequential training context of large language models. [25] also formulates KL-regularized SFT as IRL. Our work goes further by using the implicit-reward view to connect it with DPO.

### 2.2   Implicit Reward in LLMs

The typical Reinforcement Learning from Human Feedback (RLHF) methods generally involve substantial computational budgets (e.g., PPO [26] uses four models in the training process), and DPO [12] was proposed to reduce the computational overhead of the RL process. It proved that the maximization problem has a closed-form solution, allowing us to perform reward modeling directly on the implicit reward. Many related works like IPO [27], KTO [28], SimPO [29], and R-DPO [30] have continued development along this path. [17] has also established the relationship between Q-functions and LLM-logits during the DPO process. [31] treats logits as Q-values and trains a Q-head on robotic tasks. We demonstrate that analogous structures exist in the SFT process.

## 2.3 Post-Training Theory Analysis for LLMs

There are also several analyses examining relationships in post-training phases. [32] analyzes the isomorphic relationship between implicit rewards and reward models. It connects implicit rewards with the generation-verification gap. [14] pays more attention to the learning dynamics during different training phases. In our work, we focus more on the relationship between training objectives during the post-training process and use implicit rewards as a medium to understand them.

## 3 Unified View between SFT and DPO

**Token-Level MDP in LLM**  In language models, the Markov Decision Process (MDP) applies to token-level decisions. The state $s \in S$ represents the context, actions $a \in A$ are possible next tokens, and the policy $P(a|s)$ gives token probabilities. State transitions $T(s'|s,a)$ are deterministic, with $s' = s \oplus a$ (token appended to context). The model receives reward $r(s,a)$ for each choice, continuing until reaching a terminal state. $\gamma$ is the discount factor. This framework formalizes how language models make sequential decisions during text generation.

**Definition 1** (Occupancy Measure [15]). *The occupancy measure of a policy $\pi$ is defined as:*

$$\rho(s) = (1 - \gamma) \sum_{i=0}^{\infty} \gamma^i P(s_i = s|\pi),$$

*where $P(s_i = s \mid \pi)$ denotes the probability of visiting state $s$ at time step $i$ under policy $\pi$.*

**Definition 2** (State-Action Distribution [15]). *The state-action distribution of a policy $\pi$ is given by:*

$$\mu(s, a) = \pi(a|s)\rho(s),$$

*where $\mu(s, a)$ represents the stationary distribution over state-action pairs induced by $\pi$.*

### 3.1 Distribution Matching in Post-training

A well-established training objective in imitation learning is **distribution matching** [19, 15]. It focuses on minimizing the $f$-divergence between the expert's state-action distribution $\mu_E$ and that of the policy model $\mu_\pi$, while incorporating an entropy regularization term to promote exploration. However, entropy is not entirely suitable in the post-training scenario for LLMs. The use of probabilistic averaging over the whole vocabulary could potentially damage the natural language priors established during the pretraining of the base model. Therefore, we modify the regularization term from entropy to the Kullback-Leibler (KL) divergence between the base model $\pi_{ref}$ and the policy model $\pi$.

$$\min_{\pi} \ D_f(\mu_\pi \| \mu_E) \ \underbrace{+\beta D_{\mathrm{KL}}(\pi \| \pi_{ref})}_{-\beta \mathcal{H}(\pi) \text{ in traditional setting}} , \tag{1}$$

where $D_f(\cdot\|\cdot)$ denotes the $f$-divergence, and $\mathcal{H}(\pi)$ is the entropy of the policy. $\beta$ is the coefficient of the regularization term and often serves as a hyperparameter.

A similar object was also introduced in [32]. We approach this concept from a different theoretical perspective and provide additional clarification here:

1) **From the Cross-Entropy Term Perspective:** The KL divergence can be split into the entropy term and the cross-entropy term: $\mathcal{H}(\pi, \pi_{\mathrm{ref}}) - \mathcal{H}(\pi)$. As the base model has converged during the pretraining phase and has been exposed to extensive natural language data, the cross-entropy term should not be large. This implies that the policy model should not deviate from the domain of natural language when maximizing exploration.

2) **From a KL Divergence Perspective:** The base model obtained from the pretraining phase already possesses sufficient quality and contains additional knowledge. Therefore, when minimizing the divergence for distribution matching, we need to preserve the intrinsic properties of the base model.

## 3.2 Imitation Learning as Implicit Reward Discovery

Following the derivation process of non-adversarial imitation learning [15], the training objective can be expressed as an equivalent min-max problem. We have the following key result:

**Theorem 1** (Equivalent Objective for Distribution Matching). *Learning a policy that minimizes the $f$-divergence between expert and policy state-action distributions is equivalent to **first learning an optimal policy under an arbitrary reward function, then optimizing a function of that reward function**:*

$$-\min_r[\mathbb{E}_{\mu_E}[f^*(-r)] + \underbrace{\max_\pi \mathbb{E}_{\mu_\pi}[r] - \beta D_{\mathrm{KL}}(\pi\|\pi_{ref})}_{\textit{Has closed-form solution}}], \tag{2}$$

*where $f^*$ is the convex conjugate function corresponding to the chosen $f$-divergence, $\mu_\pi$ is the state-action distribution of the policy being learned, and $\mu_E$ is the expert's state-action distribution. $r$ is the independent variable of $f^*$ and is commonly interpreted as the reward function.*

We provide a detailed proof in Appendix A.1. Here, the reward function $r$ is not yet related to implicit rewards but rather represents an arbitrary function. The commonly used SFT loss still differs from distribution matching approaches in fundamental ways. However, **this formulation establishes a connection between finding a policy model and identifying a suitable reward function**. Although the equivalent objective is formulated as a bi-level optimization problem, the latter part, which we denote as $J(\pi) = \mathbb{E}_{\mu_\pi}[r] - \beta D_{\mathrm{KL}}(\pi\|\pi_{\mathrm{ref}})$, has a closed-form solution as demonstrated by [17]. We leverage this established result in our approach.

$$\pi^* = \arg\max_\pi J(\pi), \quad J(\pi^*) = V^*(s_0) \tag{3}$$

where $\pi^*(a \mid s)$ is the optimal policy. $V^*$ is the value function of the optimal policy.

**Lemma 1** (Relationship between Reward and Policy[17], **Implicit Reward**). *The relationship between reward and corresponding optimal policy is :*

$$r(x,y) = \beta \log \frac{\pi^*(y \mid x)}{\pi_{ref}(y \mid x)} + V^*(s_0) - V^*(s_t), \tag{4}$$

*where $r(x,y)$ represents the reward for the LLM's input-output pair $(x,y)$.*

As the reward $r$ in the training objective eq. (2) can be initialized arbitrarily, we can have the following assumption.

**Assumption 1** (Initial Reward Simplification). *Without loss of generality, the initial reward $r$ can be treated as $V_\pi(s_0)$ with $V_\pi(s_t) = 0$ for all $t > 0$. Under this assumption, the initial policy $\pi$ is the optimal policy with respect to the initial reward.*

Now we can substitute the latter part of eq. (2) with its closed-form solution. The correspondence between divergence measures and their conjugate functions has been established in previous work [21]. We also list them in Appendix B. We select the total variation distance as our divergence measure, for which the corresponding conjugate function is simply the identity function. Additionally, the relationship between the reward and policy $\pi$ satisfies eq. (4). This allows us to directly obtain the final objective, which takes the familiar form of SFT:

$$\max_\pi \mathbb{E}_{\mu_E}[\beta \underbrace{\log \pi(y|x)}_{\text{MLE}} - \underbrace{\log \pi_{\mathrm{ref}}(y|x) - V_\pi(s_t)}_{\text{Serve as Constants}}], \tag{5}$$

where $s_t$ represents the terminal state. The expected return after this state becomes a constant.

**Conclusion 1: Commonly used SFT is a special case of finding implicit reward, same as DPO.** We now derive that the commonly used SFT loss constitutes a *special case* of reward discovery through imitation learning when total variation is selected as the $f$-divergence measure. Appendix B presents alternative training targets derived from different $f$-divergence functions. Since the derivation process above is reversible, we conclude that the **SFT process searches along the optimal policy-reward subspace, attempting to model the reward implicitly embedded in expert demonstrations**. At the start point of the SFT process, the policy is the optimal policy under Assumption 1. During the optimization process, the relationship between the model and reward continues to satisfy eq. (4), resulting in searching the optimal subspace of policy-reward. This implicit reward structure aligns perfectly with that in DPO, offering a harmonious theoretical view that unifies both approaches.

**Conclusion 2: KL term absent in commonly used SFT.** The difference in eq. (5) between the reference model and policy model takes a zero-order form, which acts as a constant when performing stochastic gradient descent. However, it constrains the update step size of the policy model and plays an important role in most RL algorithms. **The absence of this term leads to a substantial distance between the post-DPO model's training starting point and the base model**. We propose a simple but effective method to mitigate this limitation. With smaller learning rates to reduce the optimization step size, we show significant performance improvements in the instruction following domain in Section 4.2. Furthermore, by selecting different $f$-divergences, we can derive objectives similar to SFT while preserving the KL term. Most of these involve logarithmic and exponential operations that may lead to numerical instability. We select three representative divergences and present their comparative results in Section 4.3.

### 3.3 Model Maintains Intrinsic Expected Return Estimation During SFT

In the DPO process, [17] noted that the logits of LLMs can be interpreted as a Q-function under mild assumptions. We extend this conclusion based on similar structures in eq (2).

**Theorem 2** (Intrinsic Expected Return). *During the SFT process, the logits $l_a$ of a language model correspond to the Q-function $Q(s, a)$ of the learned implicit reward:*

$$l_a = Q_{\hat{r}}(s, a) + C(s) = \hat{r}(s, a) + \gamma \mathbb{E}_{s_{t+1} \sim P(\cdot|s,a)}[V_{\hat{r}}(s_{t+1})], \tag{6}$$

*where $\hat{r}$ is the model's implicit reward function satisfying eq. (4), $\gamma$ is the discount factor, and $V_{\hat{r}}(s_{t+1})$ is the value function of the next state. $C(s)$ is a function conditioned only on the state.*

We provide a detailed proof in Appendix A.3. Our findings indicate that not only in the DPO process but also in SFT, the model's logits can be interpreted as a Q-function characterizing the model's estimation of expected returns. These returns are calculated based on **the implicit reward learned by the model itself**. The function $C(s)$ represents the gap between the true Q value and the logits, but it does not affect the relative ranking among different actions since it depends only on the current state and acts as a constant across actions. The value function can be calculated using log-sum-exp according to Appendix A, and we hypothesize that:

**Assumption 2** (Value-Dominance Assumption). *For most two states $s_1$ and $s_2$, the difference between $C(s_1)$ and $C(s_2)$ is smaller than the difference between $V(s_1)$ and $V(s_2)$.*

This conclusion allows us to use the log-sum-exp of logits from the LLM as a Value function, instead of performing Monte Carlo sampling when utilizing other divergence formats.

## 4 Empirical Study

In this section, we provide empirical analysis on the instruction-following task. Our detailed discussion is presented as follows.

- **Small learning rate during SFT process can yield significant benefits for post-DPO models.** We prove that reducing the learning rate to decrease the single-step optimization stride for SFT during sequential training improves results, as demonstrated in Section 4.2.

- **Alternative $f$-divergences that preserve KL terms also lead to better results.** Training targets derived from other $f$-divergence do not suffer from losing the KL term. We select Pearson $\chi^2$ and Squared Hellinger, which avoid numerical stability issues associated with logarithmic and exponential functions, to demonstrate these improvements in Section 4.3.

- **LLM logits exhibit value function properties, evaluating state quality similarly.** By leveraging the characteristic of value functions to reflect expected state quality, we demonstrate that different models maintain similar judgments across states in Section 4.4.

- **SFT mitigates initial reward randomness and quickly aligns implicit rewards to reasonable values.** We explain the role of SFT in post-training as correcting the initial reward Assumption 1. We demonstrate that $V(s_0)$ converges rapidly during the SFT process and present the corresponding empirical results in Section 4.5.

## 4.1 Basic Experiment Setting

**Model and Dataset Selection**   General instruction following is a fundamental capability of large language models required for most downstream tasks. Following the setting of SimPO [29], we select Llama3-8B [2] and Mistral-7B [5] as our base models. To demonstrate that our method yields consistent improvements across model scales, we additionally evaluate Qwen2.5-3B[33]. UltraChat-200K [34] is a commonly used SFT dataset. For general instruction-following tasks, models typically complete SFT training on UltraChat-200K before performing DPO on Ultra-feedback [35] to obtain the final model. We use these two datasets in our experiments.

**Hyperparameters, Device, Baselines, and Evaluation Benchmarks**   The most commonly used learning rates during post-training are 2e-5 for SFT and 5e-7 for DPO. For all of our experiments, we train with a batch size of 128 on 8×H100 GPUs using the OpenRLHF [36] framework. For the DPO training process, $\beta$ is an important hyperparameter, and we set $\beta = 0.01$. We evaluate our models using AlpacaEval2 [37], Arena Hard [38], and MT-bench [39]. Since these benchmarks can be influenced by many implementation details, such as the vLLM [40] version, we maintain consistent implementation versions with SimPO. We use the same decoding parameters as SimPO during downstream evaluation. As evaluating all three benchmarks simultaneously would incur significant API costs, in some experiments, we used AlpacaEval2 as the representative benchmark.

## 4.2 Small Learning Rate SFT Leads to Better post-DPO Results

As mentioned in Section 3.2, the KL term, i.e., $\log \pi_{\mathrm{ref}}$, that constrains the SFT learning process is a zero-order term and provides no gradient contribution to policy optimization after differentiation. Considering the importance of step size constraints in traditional RL, we infer that the learning rate for SFT should be reduced to decrease the effective update magnitude. Compared with the commonly used $2 \times 10^{-5}$, we implement smaller learning rates of $5 \times 10^{-6}$ for Llama3-Base and Qwen2.5-3B, $1 \times 10^{-6}$ for Mistral during the SFT process. For the RL process following SFT, we select DPO and SimPO algorithms while maintaining the same hyperparameters as in the $2 \times 10^{-5}$ configuration. We utilize the publicly released checkpoints from the original SimPO implementation and evaluate them in the same testing environment to establish our baseline results.

**Main Results**   Results are presented in Table 1 and Table 2. Our reproduced baseline results outperform the results reported in the original SimPO paper. We maintain identical settings to ensure fair comparison. It can be observed that reducing the learning rate leads to moderate improvements for SFT checkpoints and significant enhancements after applying alignment algorithms. The SimPO results show relative improvements of **20%** (absolute improvement of 5%) for Llama3-8B and **25%** (absolute improvement of 6%) for Mistral after applying DPO. As we maintain identical hyperparameters in the DPO training process, the performance improvements primarily derive from adjusting the learning rate during the SFT phase, which confirms our hypothesis.

Table 1: Results on Qwen2.5-3B. The other forms of SFT are compared against the result of smaller learning rate.

| Method | Qwen2.5-Base(3B) | | | | |
| | AlpacaEval 2 | | Arena-Hard | MT-Bench | |
| | LC (%) | WR (%) | WR (%) | GPT-4 Turbo | GPT-4 |
| --- | --- | --- | --- | --- | --- |
| *Baseline* | | | | | |
| SFT | 5.1 | 2.9 | 3.1 | 6.1 | 5.9 |
| + DPO | 8.8 | 6.8 | 9.9 | 7.0 | 6.9 |
| *Smaller Learning Rate SFT* | | | | | |
| SFT(smaller lr) | $5.7_{+0.6}$ | $3.3_{+0.4}$ | $3.7_{+0.6}$ | 5.9 | 5.8 |
| + DPO | $10.8_{+2}$ | $8.7_{+1.9}$ | $11.6_{+1.7}$ | 7.0 | 6.9 |
| *Other Forms of SFT* | | | | | |
| Pearson-SFT | $6.5_{+0.8}$ | $3.7_{+0.4}$ | 3.2 | $6.2_{+0.3}$ | $6.1_{+0.3}$ |
| + DPO | $11.3_{+0.5}$ | $9_{+0.3}$ | $14.3_{+2.7}$ | $7_{+0.1}$ | 6.8 |
| SH-SFT | 4.7 | 3.2 | $4.5_{+0.8}$ | $6.3_{+0.4}$ | $6.1_{+0.3}$ |
| + DPO | $11.1_{+0.3}$ | $9.1_{+0.4}$ | $13.5_{+1.9}$ | 6.9 | 6.6 |

## 4.3 Other Forms of Imitation Loss Behave Better in Sequential Training

Beyond the Total Variance divergence, other $f$-divergence functions shown in Appendix B yield training targets where the KL term is not limited to zero-order approximations. However, many alternatives involve logarithmic or exponential calculations, or even composite log-exp operations (as in Jensen-Shannon divergence), which can lead to numerical instability. We select two additional

Table 2: Downstream results for the smaller learning rate setting. The reference-only result is reported by SimPO[29]. We reproduce the SFT, DPO, and SimPO results using publicly available checkpoints. The models trained with a smaller learning rate for SFT and subsequent models fine-tuned from this SFT checkpoint are marked in `blue`.

| Method | Llama-3-Base (8B) | | | | | Mistral-Base (7B) | | | | |
|---|---|---|---|---|---|---|---|---|---|---|
| | AlpacaEval 2 | | Arena-Hard | MT-Bench | | AlpacaEval 2 | | Arena-Hard | MT-Bench | |
| | LC (%) | WR (%) | WR (%) | GPT-4 Turbo | GPT-4 | LC (%) | WR (%) | WR (%) | GPT-4 Turbo | GPT-4 |
| *Reference Only. Not Compared Directly* | | | | | | | | | | |
| RRHF[41] | 11.6 | 10.2 | 5.8 | 5.4 | 6.7 | 12.1 | 10.1 | 6.3 | 5.8 | 7.0 |
| SLiC-HF[42] | 10.9 | 8.9 | 7.3 | 5.8 | 7.4 | 12.3 | 13.7 | 6.0 | 6.3 | 7.6 |
| CPO[43] | 9.8 | 8.9 | 6.9 | 5.4 | 6.8 | 10.8 | 8.1 | 5.8 | 6.0 | 7.4 |
| IPO[27] | 11.8 | 9.4 | 7.5 | 5.5 | 7.2 | 14.4 | 14.2 | 17.8 | 6.5 | 7.4 |
| KTO[28] | 13.1 | 9.1 | 5.6 | 5.4 | 7.0 | 14.2 | 12.4 | 12.5 | 6.3 | 7.8 |
| ORPO[44] | 14.7 | 12.2 | 7.0 | 5.8 | 7.3 | 12.2 | 10.6 | 10.8 | 6.1 | 7.6 |
| R-DPO[30] | 17.4 | 12.8 | 8.0 | 5.9 | 7.4 | 17.6 | 14.4 | 17.2 | 6.6 | 7.5 |
| *Learning Rate Optimization For Supervised Fine-tuning* | | | | | | | | | | |
| SFT | 5.8 | 3.7 | 2.7 | 5.9 | 6.8 | 5.7 | 3.6 | 1.6 | 5.4 | 6.1 |
| + DPO[12] | 17.3 | 14.2 | 19.7 | 6.8 | 7.3 | 15.6 | 12.5 | 11.7 | 6.3 | 6.4 |
| + SimPO[29] | 23.5 | 21.3 | 30.3 | 7.0 | 7.3 | 24.1 | 22.9 | 22.6 | 6.5 | 6.8 |
| SFT (smaller lr) | $6.3_{+0.5}$ | $4.3_{+0.6}$ | $3.3_{+0.6}$ | 5.9 | 6.3 | $6.7_{+1.0}$ | $3.7_{+0.1}$ | $3.1_{+1.5}$ | $5.7_{+0.3}$ | $6.4_{+0.3}$ |
| + DPO[12] | $19.4_{+2.1}$ | $16.2_{+2.0}$ | $21.1_{+1.4}$ | $7.0_{+0.2}$ | $7.4_{+0.1}$ | $21.5_{+5.9}$ | $16.7_{+4.2}$ | $21.6_{+9.9}$ | $6.5_{+0.2}$ | $7.0_{+0.6}$ |
| + SimPO[29] | $28.5_{+5.0}$ | $25.0_{+3.7}$ | $34.3_{+4.0}$ | 6.6 | 7.3 | $27.3_{+3.2}$ | $24.0_{+1.1}$ | 14.9 | 6.3 | 6.8 |

$f$-divergence formulations, which are Pearson $\chi^2$ and Squared Hellinger. Their derived training targets are presented in Table 3.

For Pearson $\chi^2$, there exists a squared probability difference term that acts as a KL constraint. For Squared Hellinger, a coefficient term related to probability differences is multiplied before the classic gradient term after applying the chain rule, which modulates the update step size. We compare these results with our previously obtained Total Variance results.

Table 3: We list three optional $f$-divergences and their corresponding training targets. We use $\Delta V$ as an abbreviation for $V_\pi(s_0) - V_\pi(s_t)$ for simplicity of notation. See Appendix B for the complete table.

| Name | $D_f(P\|Q)$ | Conjugate $f^*(t)$ | Training Target $\max_\pi \mathbb{E}_{\mu_E}[\cdot]$ |
|---|---|---|---|
| Total variation | $\frac{1}{2}\int |p(x) - q(x)|\,\mathrm{d}x$ | $t$ | $\beta \log \pi(y|x) - \log \pi_{\text{ref}}(y|x) - V_\pi(s_t)$ |
| Pearson $\chi^2$ | $\int \frac{(q(x)-p(x))^2}{p(x)}\,\mathrm{d}x$ | $\frac{1}{4}t^2 + t$ | $-\frac{1}{4}(\log \frac{\pi(y|x)}{\pi_{ref}(y|x)} + \Delta V)^2 + \log \frac{\pi(y|x)}{\pi_{ref}(y|x)} + \Delta V$ |
| Squared Hellinger | $\int \left(\sqrt{p(x)} - \sqrt{q(x)}\right)^2 \mathrm{d}x$ | $\frac{t}{1-t}$ | $1 - \frac{1}{1+\log \frac{\pi(y|x)}{\pi_{ref}(y|x)}+\Delta V}$ |

Table 4: Downstream results for different training targets and their corresponding post-DPO checkpoints. SFT refers to the commonly used training target derived from total variation. Pearson-SFT refers to the imitation objective derived from Pearson-$\chi^2$ divergence. SH-SFT refers to the objective derived from Squared Hellinger divergence.

| Method | Llama-3-Base (8B) | | | | | Mistral-Base (7B) | | | | |
|---|---|---|---|---|---|---|---|---|---|---|
| | AlpacaEval 2 | | Arena-Hard | MT-Bench | | AlpacaEval 2 | | Arena-Hard | MT-Bench | |
| | LC (%) | WR (%) | WR (%) | GPT-4 Turbo | GPT-4 | LC (%) | WR (%) | WR (%) | GPT-4 Turbo | GPT-4 |
| *Traditional SFT* | | | | | | | | | | |
| SFT | 6.3 | 4.3 | 3.3 | 5.9 | 6.3 | 6.7 | 3.7 | 3.1 | 5.7 | 6.4 |
| + DPO | 19.4 | 16.2 | 21.1 | 7.0 | 7.4 | 21.5 | 16.7 | 21.6 | 6.5 | 7.0 |
| *Other Format of SFT* | | | | | | | | | | |
| Pearson-SFT | 5.1 | 3.7 | 3.3 | 5.8 | 6.1 | 6.2 | 3.4 | 3.5 | 6 | 6.5 |
| + DPO | $20.1_{+0.7}$ | $17.7_{+1.5}$ | $24.7_{+3.6}$ | $7.1_{+0.1}$ | 7.2 | $23.1_{+1.6}$ | $19.0_{+2.3}$ | $21.8_{+0.2}$ | $6.8_{+0.3}$ | 6.7 |
| SH-SFT | 4.8 | 3.7 | 17.3 | 5.7 | 6 | 6.5 | 3.5 | 3.1 | 5.7 | 6.1 |
| + DPO | $19.6_{+0.2}$ | $17.3_{+1.1}$ | 19.9 | 6.9 | 7.2 | $23.6_{+2.1}$ | $20.9_{+4.2}$ | $22_{+0.4}$ | $6.8_{+0.3}$ | 6 |

**Main result** The results are presented in Table 4. It can be observed that both the Pearson $\chi^2$ and Squared Hellinger lead to weaker SFT but better results after DPO, regardless of whether we

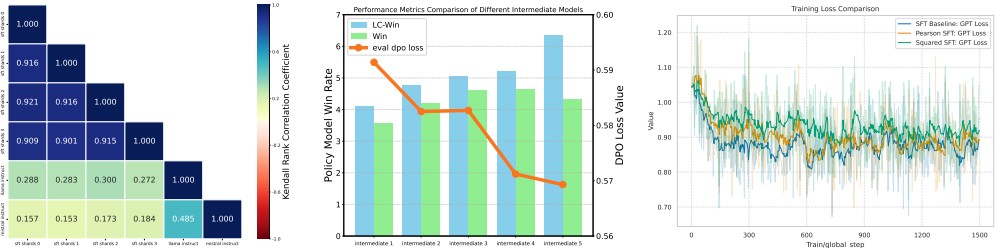

(a) KLCC between different checkpoints

(b) Alpaca-Eval 2 results and corresponding DPO Loss.

(c) Training loss for different objectives.

Figure 2: **Left**: We obtain rankings of different models on identical step-wise instruction-response pairs and calculate KLCC to measure agreement between these rankings. **Middle**: Lower values of DPO Loss indicate better alignment between implicit reward and environment reward. And the corresponding AlpacaEval2 result. **Right**: Training loss for objectives derived from: total variation, Pearson $\chi^2$, and squared Hellinger.

use Mistral or Llama. We can reach an interesting conclusion that a better SFT checkpoint doesn't necessarily lead to better DPO results. These improvements from KL-regularized SFT validate our theory, showing the importance of the KL term during post-training. The training loss curves for these three SFT approaches are shown in Figure 2c.

## 4.4 Value and Reward in SFT

With the theoretical conclusions presented in Section 3.3, we estimate the value function using the logits of LLMs. Traditional value estimation typically involves Monte Carlo sampling. Moreover, values are conventionally calculated using ground truth rewards, which in our context are implicit and not directly accessible. We aim to provide empirical evidence that LLM logits exhibit properties of value functions: their scores can be used to evaluate state quality.

More precisely, we demonstrate that for LLMs trained on the same domain, the evaluations for different states maintain similar rankings across models. We divide the UltraChat-200k dataset into 4 splits and perform SFT on Llama-3-base to obtain 4 different checkpoints. We also select Llama-3-instruct as a representative model that shares the same prior but was trained on different datasets, and Zephyr [45] as a model with a different prior but trained on similar datasets. For clearly defined steps, we choose MATH-500 [46] as the validation set, sample one trajectory for each question, and split the reasoning path into steps. We extract the logits from the model's output after inputting the final token of each step, calculate the log-sum-exp, and rank these values within each individual model. If the logits of an LLM possess the property of evaluating state quality, they should share similar ranks across models. We calculate the Kendall rank correlation coefficient (KLCC).

**Main Results**  The results are shown in Fig. 2a. We find that the value rankings of LLMs are positively correlated across all experimental settings. The correlation approaches 1 for the four dataset shards and remains positive even when the post-training dataset or model prior changes. An interesting observation is that the ranking correlation between Zephyr and Llama3-instruct is significantly higher than that between Zephyr and our sharded models, despite Zephyr being trained on UltraChat rather than the same dataset. The positive correlation indirectly validates that the logits function as a value, thus confirming our Assumption 2.

**Another observation for implicit reward**  We find that stronger alignment between the model's implicit reward and downstream reward correlates with better model performance. During the SFT training process, we calculate the DPO loss on previously annotated pairwise data of AlpacaEval 2 used as an evaluation set. As the benchmark serves as an environment, lower DPO loss indicates greater consistency between implicit rewards and the environment. The results are shown in Figure 2b. We observe that alignment between rewards and the environment positively correlates with model performance, which aligns with intuitive expectations.

## 4.5  Reward Stabilization through SFT

SFT in our theory is attempting to bring the implicit reward into an appropriate range for fine-grained modifications. Assumption 1 assumes that at the starting point of the implicit reward search process, the model represents the optimal policy under some unknown reward function. This reward function may significantly differ from the reward function in real downstream tasks. We plot the training curve of the log-sum-exp of the first logits after the prompt and create some early-exit SFT checkpoints to perform DPO on them, as shown in Figure 3. It can be observed that the $V(s_0) = \log \Sigma(\cdot)$ increases rapidly and converges quickly. The downstream task performance exhibits highly consistent trends. We conclude that SFT has already completed its task of bringing the implicit reward to a reasonable range by 150 steps, and subsequent steps focus on more refined modeling.

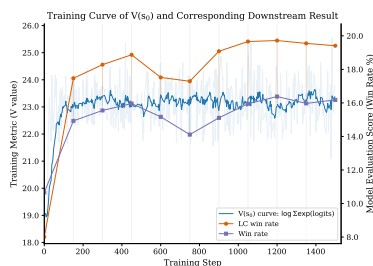

Figure 3: The trend of $V(s_0)$ using the logits of the first response token. Performance on AlpacaEval 2 for post-DPO on SFT checkpoints at corresponding training steps.

# 5  Broader Impacts, Limitations and Future Work

**Broader Impacts: Philosophical Dimension**  As we mentioned, models learn implicit rewards during SFT, which may lead to further discussions about whether LLMs can be considered entities with preset environmental awareness. This opens philosophical inquiries into the nature of consciousness and the extent to which artificial systems might exhibit consciousness-like properties.

**Limitation: We did not experimentally explore additional divergence functions**  The commonly used KL divergence and JS divergence involve logarithmic and exponential calculations, which can lead to numerical instability. We have attempted various implementations and performed small-value clipping on the data. Although this prevented NaN errors, the loss would still reach extremely large values, such as 6e7. We did not design specialized operators to implement these methods, resulting in unknown effectiveness for these KL divergence approaches.

**Future Work: SFT and DPO multi-object learning**  Since both the SFT process and DPO process model the implicit reward, a natural idea is to formulate them into multi-objective learning rather than sequential training. Appendix C details some failed attempts at implementing this multi-objective approach, hoping these findings can contribute to future research in this direction.

# 6  Conclusion

In conclusion, we establish implicit reward learning as a unifying view connecting SFT and preference learning in LLM post-training. We demonstrate that conventional SFT is a special case of implicit reward learning using total variation divergence, limited by an absent KL term. Our approach of reducing learning rates significantly improves model performance, while alternative $f$-divergence objectives preserving the KL term show additional gains. We extend DPO's logits-to-Q-function mapping to SFT and confirm SFT's crucial role in stabilizing random implicit rewards, advancing both theoretical understanding and practical strategies for more effective post-training.

## Acknowledgements

This work was supported by the National Natural Science Foundation of China (No. U24B20181) and Shanghai Pilot Program for Basic Research - Fudan University 21TQ1400100 (22TQ018).

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

# A  Theoretical Proofs

In this section, we provide detailed proofs of our theoretical results.

**Lemma 2** (Fixed-Point Solution for Maximum-Entropy RL [17])**.** *The optimal policy $\pi^*(a_t \mid s_t)$ and the corresponding optimal value function $V^*(s_t)$ in the maximum-entropy framework satisfy the following fixed-point equations:*

$$\pi^*(a_t \mid s_t) = \exp\left(\frac{Q^*(s_t, a_t) - V^*(s_t)}{\beta}\right), \tag{7}$$

$$V^*(s_t) = \beta \log \int_{\mathcal{A}} \exp\left(\frac{Q^*(s_t, a_t)}{\beta}\right) da_t, \tag{8}$$

$$J(\pi^*) = V^*(s_0), \tag{9}$$

*where $\pi^*(a \mid s)$ is the optimal policy. $Q^*$ and $V^*$ are the quality function and value function of the optimal policy.*

## A.1  Non-adversarial Imitation: Finding Reward

**Theorem 3** (Equivalent Objective for Distribution Matching)**.** *Learning a policy that minimizes the $f$-divergence between expert and policy state-action distributions is equivalent to **first learning an optimal policy under an arbitrary reward function, then optimizing a function of that reward function**:*

$$\min_{\pi} \ D_f(\mu_\pi \| \mu_E) + \beta D_{\mathrm{KL}}(\pi \| \pi_{ref}), \tag{10}$$

$$= -\min_{r} \mathbb{E}_{\mu_E}[f^*(-r)] + \underbrace{\max_{\pi} \mathbb{E}_{\mu_\pi}[r] - \beta D_{\mathrm{KL}}(\pi \| \pi_{ref})}_{\text{Has closed-form solution}}, \tag{11}$$

*where $f^*$ is the convex conjugate function corresponding to the chosen $f$-divergence, $\mu_\pi$ is the state-action distribution of the policy being learned, and $\mu_E$ is the expert's state-action distribution.*

*Proof.* Following the derivation process of non-adversarial distribution matching [16], the training objective 1 can be rewritten into a min-max problem with a conjugate function:

$$\text{Given that } f^*(g) = \max_{x \in \mathrm{dom}(f)} \{xg - f(x)\}, \quad f(x) = \max_{g \in \mathrm{dom}(f^*)} \{xg - f^*(g)\}, \tag{12}$$

$$\tag{13}$$

$$\min_{\pi} D_f(\mu_\pi \| \mu_E) + \beta D_{\mathrm{KL}}(\pi \| \pi_{\mathrm{ref}}), \tag{14}$$

$$= \min_{\pi} \int \mu_E f\left(\frac{\mu_\pi}{\mu_E}\right) ds da + \beta D_{\mathrm{KL}}(\pi \| \pi_{\mathrm{ref}}), \tag{15}$$

$$= \min_{\pi} \int \mu_E \left(\max_{g:S \times A \rightarrow \mathrm{dom}(f^*)} \{\frac{\mu_\pi}{\mu_E} g - f^*(g)\}\right) ds da + \beta D_{\mathrm{KL}}(\pi \| \pi_{\mathrm{ref}}), \tag{16}$$

$$= \min_{\pi} \max_{g:S \times A \rightarrow \mathrm{dom}_{f*}} -\mathbb{E}_{\mu_E}[f^*(g)] + \mathbb{E}_{\mu_\pi}[g] + \beta D_{\mathrm{KL}}(\pi \| \pi_{\mathrm{ref}}). \tag{17}$$

$\square$

By substituting $g$ with $-r$, we can rewrite the min-max formulation:

$$-\max_{\pi} \min_{r:S \times A \rightarrow -\mathrm{dom}_{f*}} \mathbb{E}_{\mu_E}[f^*(-r)] + \mathbb{E}_{\mu_\pi}[r] - \beta D_{\mathrm{KL}}(\pi \| \pi_{\mathrm{ref}}). \tag{18}$$

The target is also a saddle point problem, as the conjugate function exhibits convexity and the KL term can be split into an entropy term that is concave and a cross-entropy term that is linear with respect to $\pi$. Therefore, the order of min-max operations can be exchanged:

$$-\min_{r:S \times A \rightarrow -\mathrm{dom}_{f*}} \max_{\pi} \mathbb{E}_{\mu_E}[f^*(-r)] + \mathbb{E}_{\mu_\pi}[r] - \beta D_{\mathrm{KL}}(\pi \| \pi_{\mathrm{ref}}). \tag{19}$$

The maximization is only performed on the latter part of the expression and has a closed-form solution according to previous work [17]:

$$-\min_{r} \mathbb{E}_{\mu_E}[f^*(-r)] + \underbrace{\max_{\pi} \mathbb{E}_{\mu_\pi}[r] - \beta D_{\mathrm{KL}}(\pi \| \pi_{\mathrm{ref}})}_{\text{Has closed-form solution}}, \tag{20}$$

## A.2 SFT is a Special Case

The final objective is

$$- \min_r [\mathbb{E}_{\mu_E}[f^*(-r)] + \underbrace{\max_\pi \mathbb{E}_{\mu_\pi}[r] - \beta D_{\mathrm{KL}}(\pi \| \pi_{\mathrm{ref}})}_{\text{Has closed-form solution}}], \tag{21}$$

$$= - \min_r \mathbb{E}_{\mu_E}[f^*(-r)] + V_\pi(s_0). \tag{22}$$

Throughout the training process, the reward maintains the equation (4) and the policy $\pi$ is always the optimal solution of the latter parts. When choosing the Total Variation distance, the conjugate function is $f^*(t) = t$, and the training objective is just the MLE term shown below:

$$- \min_r \mathbb{E}_{\mu_E}[f^*(-r)] + V_\pi(s_0), \tag{23}$$

$$= \max_r \mathbb{E}_{\mu_E}[-f^*(-r)] - V_\pi(s_0), \tag{24}$$

$$= \max_r \mathbb{E}_{\mu_E}[r] - V_\pi(s_0), \tag{25}$$

$$= \max_r \mathbb{E}_{\mu_E}[\log \frac{\pi(y|x)}{\pi_{\mathrm{ref}}(y|x)} + V_\pi(s_0) - V_\pi(s_t)] - V_\pi(s_0), \tag{26}$$

$$= \max_\pi \mathbb{E}_{\mu_E}[\beta \underbrace{\log \pi(y|x)}_{\text{MLE}} - \underbrace{\log \pi_{\mathrm{ref}}(y|x)}_{\text{constant}} - V_\pi(s_t)]. \tag{27}$$

## A.3 SFT Maintains Intrinsic Expected Return Estimation

**Theorem 4** (Intrinsic Expected Return). *During the SFT process, the logits $l_a$ of a language model correspond to the Q-function $Q(s, a)$ of the learned implicit reward:*

$$l_a = Q_{\hat{r}}(s, a) + C(s) = \hat{r}(s, a) + \gamma \mathbb{E}_{s_{t+1} \sim P(\cdot|s,a)}[V_{\hat{r}}(s_{t+1})], \tag{28}$$

*where $\hat{r}$ is the model's implicit reward function satisfying eq. (4), $\gamma$ is the discount factor, and $V_{\hat{r}}(s_{t+1})$ is the value function of the next state. $C(s)$ is a function conditioned only on the state.*

*Proof.* The language model represents token probabilities through a softmax operation over logits $l$:

$$p(a_i|s) = \frac{e^{l_i/\tau}}{\sum_j e^{l_j/\tau}}, \tag{29}$$

where $\tau$ is the temperature parameter, typically set to 1 during training.

As previously discussed, the model represents the optimal policy under some implicit reward function. The probability distribution for this optimal policy satisfies:

$$p(a_i|s) = \frac{e^{Q(s,a_i)/\beta}}{\sum_j e^{Q(s,a_j)/\beta}}. \tag{30}$$

Equating these expressions, we have:

$$\frac{e^{l_i/\tau}}{\sum_j e^{l_j/\tau}} = \frac{e^{Q(s,a_i)/\beta}}{\sum_j e^{Q(s,a_j)/\beta}}, \tag{31}$$

$$e^{l_i/\tau} = e^{Q(s,a_i)/\beta} \cdot \frac{\sum_j e^{l_j/\tau}}{\sum_j e^{Q(s,a_j)/\beta}}, \tag{32}$$

$$e^{l_i/\tau} = k \cdot e^{Q(s,a_i)/\beta}, \tag{33}$$

$$l_i = \frac{\tau}{\beta} Q(s, a_i) + C(s), \tag{34}$$

where $\beta$ is the KL divergence coefficient. The relationship shows that logits $l_i$ have a linear mapping to the Q-values, with $C(s)$ depending only on the state since it incorporates terms summed over all possible actions. We hypothesize that $C(s)$ remains numerically similar across different states, and we provide supporting evidence for this in the experimental section. □

# B Different f-Divergence Leads to Different Loss Format

In this section, we present the detailed formulation of loss functions derived from different $f$-divergences.

| Name | $D_f(P\|Q)$ | Conjugate $f^*(t)$ | Training Target |
|---|---|---|---|
| | | | $\max_\pi \mathbb{E}_{\mu_E}[\cdot]$ |
| Total variation | $\frac{1}{2}\int |p(x)-q(x)|\,\mathrm{d}x$ | $t$ | $\beta\log\pi(y|x) -$ $\log\pi_{\text{ref}}(y|x) - V_\pi(s_t)$ |
| Kullback-Leibler (KL) | $\int p(x)\log\frac{p(x)}{q(x)}\,\mathrm{d}x$ | $\exp(t-1)$ | $[-\pi_{ref}(y|x)+\pi(y|x)]\cdot$ $\exp(-\Delta V - 1) - V_\pi(s_0)$ |
| Reverse KL | $\int q(x)\log\frac{q(x)}{p(x)}\,\mathrm{d}x$ | $-1-\log(-t)$ | $1+\log(\log\frac{\pi(y|x)}{\pi_{ref}(y|x)}+\Delta V)$ $-V_\pi(s_0)$ |
| Pearson $\chi^2$ | $\int\frac{(q(x)-p(x))^2}{p(x)}\,\mathrm{d}x$ | $\frac{1}{4}t^2+t$ | $-\frac{1}{4}(\log\frac{\pi(y|x)}{\pi_{ref}(y|x)}+\Delta V)^2$ $+\log\frac{\pi(y|x)}{\pi_{ref}(y|x)}+\Delta V$ |
| Squared Hellinger | $\int\left(\sqrt{p(x)}-\sqrt{q(x)}\right)^2\mathrm{d}x$ | $\frac{t}{1-t}$ | $1-\frac{1}{1+\log\frac{\pi(y|x)}{\pi_{ref}(y|x)}+\Delta V}$ |
| Jensen-Shannon | $\frac{1}{2}\int p(x)\log\frac{2p(x)}{p(x)+q(x)}+$ $q(x)\log\frac{2q(x)}{p(x)+q(x)}\,\mathrm{d}x$ | $-\log(2-\exp(t))$ | $\log(2-\frac{\pi_{ref}(y|x)}{\pi(y|x)}\cdot$ $\exp(-\Delta V))$ |
| GAN | $\int p(x)\log\frac{2p(x)}{p(x)+q(x)}+$ $q(x)\log\frac{2q(x)}{p(x)+q(x)}\,\mathrm{d}x-\log(4)$ | $-\log(1-\exp(t))$ | $\log(1-\frac{\pi_{ref}(y|x)}{\pi(y|x)}\cdot$ $\exp(-\Delta V))$ |

Table 5: For different $f$-divergences, we list their corresponding conjugate functions $f^*$ and the derived training targets. We use the symbol $\Delta V$ as an abbreviation for $V_\pi(s_0) - V_\pi(s_t)$ for simplicity of notation.

# C Multi-Object

Since both the SFT process and DPO process model the implicit reward, we investigate whether they can be formulated as multi-objective learning rather than sequential training. We approach this using Lagrangian methods. Specifically, we formulate the multi-objective learning process using the Lagrangian multiplier method. We assume that during the SFT process, the accuracy of the DPO loss should also be kept as low as possible. We formulate this with the following objective:

$$\min_\pi \text{SFT loss, s.t., (accuracy of (chosen, reject)} - \text{target acc)} < \delta, \tag{35}$$

$$\min_\pi \text{SFT loss} + \lambda \cdot \text{DPO loss}. \tag{36}$$

Where $\delta$ is the hyperparameter. Following the process of PPO, the parameter $\lambda$ is adjusted to $\frac{\lambda}{2}$, when $\text{acc} - \text{target acc} < \delta$ and is set to $2 \cdot \lambda$ otherwise.
We train the model directly on the UltraFeedback dataset, and some training metrics during one training are shown below.

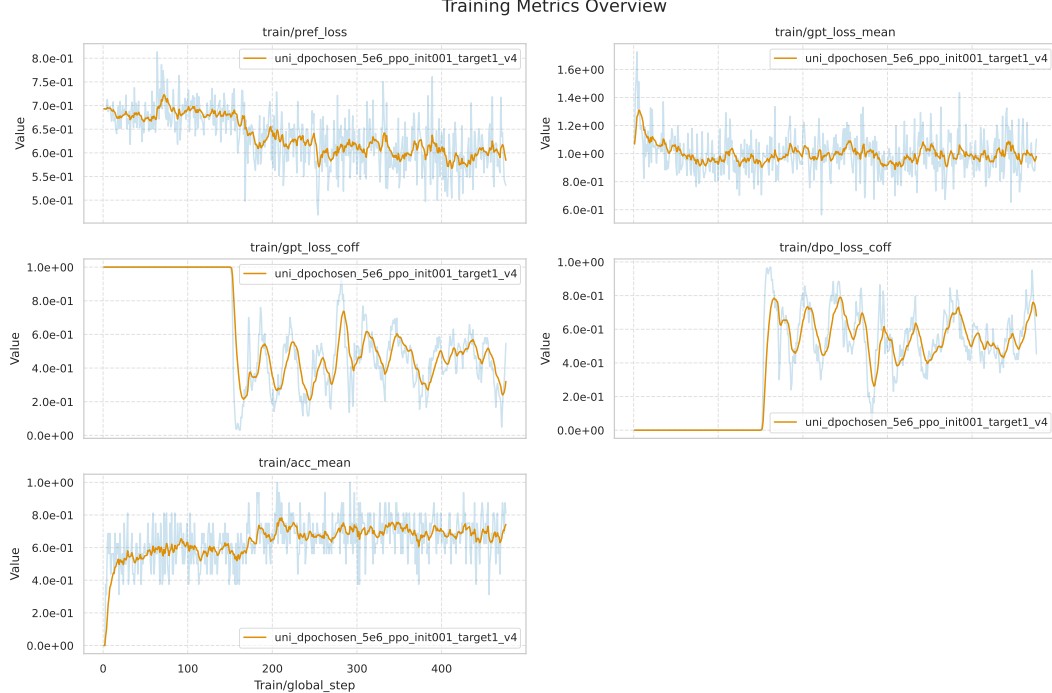

Figure 4: Training Metric Visualization for Multi-object Learning

We also experiment with various hyperparameters, including different growth and decay coefficients for $\lambda$, but all of these configurations lead to poor downstream results. We think that our failure appears consistent with the theory presented in [14]. As the SFT loss coefficient increases, it tends to increase probability mass in negative regions, thus diminishing the effectiveness of the squeezing effect. We validate in other experiments through interleaved experiments, where we split the SFT and DPO data into four segments and train them in an interleaved manner. The results are shown below.

Table 6: Results for interleaved SFT and DPO training. The table shows the performance after each training stage in terms of Win Rate and LC Win Rate percentages.

| Training Stage | AlpacaEval 2 | |
|---|---|---|
| | Win (%) | LC-Win (%) |
| *Interleaved Training* | | |
| SFT 1 | 5.28 | 4.86 |
| DPO 1 | 7.03 | 8.11 |
| SFT 2 | 5.06 | 4.63 |
| DPO 2 | 11.27 | 11.94 |
| SFT 3 | 8.94 | 4.83 |
| DPO 3 | 12.94 | 13.30 |
| SFT 4 | 8.44 | 5.13 |
| DPO 4 | 8.24 | 5.02 |

It can be observed that after each DPO stage, subsequent SFT training experiences a significant performance drop, indicating that SFT and DPO objectives are inherently conflicting in our experimental setting.

However, our failure does not necessarily mean this approach is fundamentally infeasible. We have not yet explored its effectiveness in other domains, and we leave this investigation to future work. We hypothesize that in tasks where expert trajectories more closely align with SFT's assumption that the training data represents optimal behavior, the results may be more promising.

