# OpenReview forum: "Implicit Reward as the Bridge: A Unified View of SFT and DPO Connections"
_NeurIPS.cc/2025/Conference — NeurIPS 2025 poster_

### Official Review · Reviewer_6WqT · 2025-06-16

**Clarity:** 4
**Significance:** 4
**Originality:** 4
**Rating:** 5
**Confidence:** 5

**Summary:**

Previously, [12,17] have established the relationship between DPO and implicit reward learning and how LLM logits can function as a Q-function with regard to that implicit reward model. This paper extends this relationship to SFT, and discovered that KL term is irrelevant to the policy in its mathematical form.

A learning rate adjustment scheme is proposed and the KL div is extended to tother f-divs.

**Questions:**

- L46, do you mean Figure 4 or Figure 1? Elsewhere, Figure 1 is never mentioned. Thus I wonder how it is made?
- Figure 2 a, consider flipping y axis and make image texts larger?

**Ethical Concerns:**

["NO or VERY MINOR ethics concerns only"]

**Quality:**

4

**Strengths And Weaknesses:**

Strength:

- This paper offers deep theoretical insights, filling an important gap in previous analysis in RLHF.
- The insight gleaned from theoretical analysis is converted into a learning rate tuning insight which turns out well empirically. The f-div SFT experiments showed weaker results.
- Various observational experiments show convincing results on the theoretical insights.

Weakness:

- The multi-objective results in Appendix D provides empirical results that joint optimization fails, revealing a practical inconsistency that undermines the central theoretical claim. It should be considered that more efforts be spent on looking at this.

---

> ### Author Rebuttal · Authors · 2025-07-29
>
> We are very grateful for your recognition and encouragement of our work. We are deeply touched. Regarding your two questions:
>
> ## Typo in L46
>
> There is indeed a typo on line 46. It should be Figure 1. We made an error when referencing the figure, which caused confusion in understanding. We sincerely apologize.
> ## About Figure suggestion
>
> Regarding 2a, you provided a very sincere suggestion. We will correct the figure according to your recommendation in the subsequent version. We apologize that the figure is too small, causing reading inconvenience.
>
> ## About the failure of joint optimization
>
> We also feel it is very regrettable. There are many possible reasons that lead to the failure of joint optimization. We think the problem mainly lies in data quality. We will dive into this question further and try to solve this in future work.
>
> Thank you again for your support and recognition of our work!

---

### Official Review · Reviewer_SPrv · 2025-07-01

**Clarity:** 3
**Significance:** 3
**Originality:** 3
**Rating:** 3
**Confidence:** 4

**Summary:**

This paper presents a unified theoretical framework that connects Supervised Fine-Tuning and Direct Preference Optimization in the post-training of Large Language Models. The authors argue that both SFT and DPO can be viewed as learning under an implicit reward function within the same policy-reward subspace. They derive the SFT objective from f-divergence-based distribution matching and show it is a special case of implicit reward learning when total variation is used. Based on this insight, the paper proposes two practical improvements: (1) reducing the learning rate during SFT to compensate for the loss of gradient information from the KL term, and (2) using alternative f-divergence formulations that preserve KL structure. Empirical evaluations on standard instruction-following benchmarks demonstrate that these simple changes sometimes improve post-DPO performance.

**Questions:**

Q1: Why do the benefits of different SFT variants only emerge after post-alignment training? The paper observes that certain SFT configurations perform better after DPO but not immediately after SFT. Are there theoretical or empirical insights into why this discrepancy exists? Why would SFT effectiveness not manifest directly in post-SFT performance?

Q2: Line 178–180: The authors argue that omitting the KL term leads to a “substantial distance” between the policy and reference model, implying this is undesirable. However, prior works have shown that SFT alone is effective and a larger distance may offer greater expressivity. Why is substantial distance necessarily a negative factor in this case?

**Ethical Concerns:**

["NO or VERY MINOR ethics concerns only"]

**Final Justification:**

I find the theoretical framework of this paper interesting. The authors addressed some of my questions, but most of the experiment-related concerns remain unaddressed. However, I still have doubts about the practical value of the claim that “a smaller learning rate generally leads to better performance.” Although the authors added a new model, I do not see its results on "Other Imitation Loss" experiments. Overall, I still consider the practical applicability of this work to be limited, and the current experiments are insufficient to fully support its theoretical framework (align with reviewer 4L4H). Therefore, I will keep my score unchanged

**Limitations:**

yes

**Quality:**

2

**Strengths And Weaknesses:**

## Strength.

- **Theoretical contribution.** The paper offers a rigorous and insightful theoretical framework that unifies SFT and DPO through the lens of implicit reward learning. This clarifies an often empirically observed but poorly understood connection.
- **Practical utility.** The proposed improvements are easy to implement and demonstrate consistent gains across different models and datasets, making them applicable in real-world settings.

---

## Weaknesses.

- **W1: Insufficient justification for selected learning rate**

In Section 4.2, the authors claim that using a smaller learning rate during SFT leads to better post-DPO performance. However, the choice of 5e-6 as the "small" learning rate seems arbitrary. A more thorough hyperparameter study would be helpful to validate this claim. Would further reducing the learning rate continue to improve results, or is there an optimal range?

- **W2: Limited model evaluation**

All experiments are conducted on 7B-level models. It remains unclear whether the observed phenomenon generalizes across smaller (e.g., 0.5B, 1.5B, 3B) or larger (e.g., 13B) models. Additionally, the authors did not experiment with state-of-the-art open-source models such as the Qwen series, which may yield more robust or contrasting results.


- **W3: Lack of discussion on Table 3**

 Table 3 contains several outlier values, and the proposed SFT variants show inconsistent performance across benchmarks. However, the authors provide no analysis or explanation for these inconsistencies, leaving unclear when and why the proposed methods succeed or fail.

- **W4: No representation-level analysis**

The claims regarding KL divergence and its effect on alignment could be further substantiated by analyzing changes in the learned representation space. For instance, examining whether the regularization leads to more structured or aligned latent spaces would strengthen the argument.


- **W5: Limited generalizability beyond DPO**

The paper does not provide sufficient analysis on the sensitivity of key hyperparameters, such as the clipping bounds in pre-clip, the threshold in advantage filtering, number of training data samples, or the weights in the final reward aggregation. Without such studies, it is difficult to assess the robustness and reproducibility of the proposed StableReinforce algorithm.

- **W6: Presentation issue**

The reference to a figure on line 45 appears incorrect.

---

> ### Author Rebuttal · Authors · 2025-07-29
>
> Thank you for your kind suggestions.
> ## About learning rate selection
> Regarding the first point in the weaknesses, we also tried other learning rates. For the Llama model, with a learning rate of 1e-6, the DPO results were 18.83 and 17.26, which are also better than the commonly used 2e-5 learning rate. Our SFT learning rate adjustment is a trick based on theoretical properties, and the results with other learning rates on Llama hopefully demonstrate the universality of small learning rates for performance improvement.
> ## About model selection
> We followed SimPO's experimental settings to facilitate using their results as baseline comparisons. SimPO uses exactly the same settings as ours, and such model configurations are very common in preference learning, as can be seen in SimPO, DPO, ORPO, and GEM. So, maybe this setting is enough?
> ## About SFT and DPO relationship
> Our work mainly focuses on the connection between SFT and DPO. As we mentioned in the experimental section L268, a good SFT starting point may not necessarily score high on benchmarks, so SFT performance has variability, but we achieved significant improvements on algorithms like DPO and SimPO.
> ## About representation-level analysis
> In Section 4.4, our analysis of logits changes, and in Section 4.5, our visualization of logits value changes during the SFT process, are both demonstrations of what the model is doing at the representation level.
> ## About Key parameters
> Our work does not involve the key parameters you mentioned, such as clipping bounds in pre-clip and the threshold in advantage filtering. Your question has left us very confused.
> ## About Typo in Line 45
> The figure reference on line 45 is incorrect, and we sincerely apologize.
> ## About why increments emerge after post-alignment training?
> SFT mainly provides a better initial point for DPO in the post-training process. As we limited the update magnitude during the SFT stage via kl or a small learning rate to ensure the exploratory nature of the SFT checkpoint, it may not perform well on specific downstream tasks. However, it maintains output diversity and acts as a better starting point for alignment algorithms, which contributes to better results after performing alignment algorithms.
> ## About why KL is suitable
> The distance relationship between the SFT model and the base model is also relative. When LLM pre-training converges, the cross-entropy loss is around 1. However, for the original maximum entropy, taking Qwen as an example, the entire vocabulary size is 128,256, so the maximum entropy at uniform distribution is log(1/128,256) = 5.11. It is significantly larger than the cross-entropy loss. When the maximum entropy is achieved, it completely loses the characteristics of language distribution. Therefore, although the SFT stage changes output probabilities, as long as the output is natural language, the cross-entropy loss constraint is reasonable. We constrain the relationship with the base model to maintain the rich natural language modeling in the base model.

---

> > ### Comment · Reviewer_SPrv · 2025-08-06
> >
> > Thank you for your response.
> >
> > First, I apologize for the confusion caused by my comment on W5. I accidentally pasted a reply intended for another paper without double-checking, which resulted in a mismatch between the title and the content. As indicated in the title of W5, what I actually intended to point out was whether the conclusions of the paper hold for other variants of DPO. Specifically, in Table 1, do the conclusions still hold for methods other than DPO and SimPO? In Table 3, do they hold for methods other than DPO?
> >
> > Regarding W1, I am a bit confused about for which benchmarks the results of 18.83 and 17.26 are reported? While these results are better than the commonly used learning rate of 2e-5, are they also better than the results obtained with the 5e-6 learning rate used in your paper? If not, why does a smaller learning rate not lead to better results?
> >
> > For W2, you explained that you did not experiment on more models because you followed the settings of SimPO, DPO, ORPO, and GEM. I have checked these papers. SimPO also conducted experiments on Gemma2-9B, while ORPO’s model selection covers three model families (Phi-2, Llama-2, and Mistral) and two different model sizes (2.7B and 7B). Could you please provide the full name or reference for GEM? I still question whether the current experimental coverage is sufficient.
> >
> > Regarding W3, I was originally referring to the results after DPO and SimPO. It can be observed that on some benchmarks, performance decreases. The paper does not provide an analysis or discussion of these cases, and only states that these results “lead to better results after DPO.”

---

> > > ### Author Response · Authors · 2025-08-07
> > > **Further Discussion**
> > >
> > > We sincerely apologize for not providing a comprehensive reply in our initial response. Due to an error with W5, your initial review comments appeared to be very confusing,  even suspected they might be AI-generated. We deeply regret this situation. Thank you for your willingness to engage in this discussion. Here are our further responses.
> > >
> > > ## Before we discuss the experiment
> > >
> > > Most of your concerns are related to our experiments. We would like to first reiterate that our core contribution is providing a unified theoretical perspective on SFT and DPO. Within this framework, we identified issues with existing training paradigms and conducted some $\textbf{necessary}$ experiments to further validate our theoretical contributions
> > >
> > > ## About Learning rate selection
> > >
> > > We take your concern about the learning rate very seriously and have supplemented our results with experiments using 1e-6. These additional results were evaluated on Alpaca-Eval 2 and demonstrate performance superior to the original results.
> > >
> > > Given that learning rate serves as an optional solution for compensating KL divergence issues—similar to how the beta coefficient affects KL in DPO experiments, where a larger beta doesn't necessarily yield better results, but adding KL constraints generally improves performance—we do not claim that smaller learning rates always lead to better performance. Rather, we aim to demonstrate that smaller learning rates are generally a better choice, and we have elaborated on the theoretical reasoning in our theoretical section.
> > >
> > > ## About Model Selection
> > >
> > > In our previous response, we mentioned several related works:
> > >
> > > - SimPO's experimental setting: SimPO has released three versions of their paper. The Gemma-2 experiments were added in the third version, which was released much later after their initial work to demonstrate the method's effectiveness. We followed their first two early versions, in which they used results identical to ours to demonstrate the effectiveness of their approach.
> > > - ORPO: While ORPO conducted more extensive experiments on the phi-2.7B model, we intended to reference their highlighted experimental results. The main results presented in Figure 1 of their paper were on Llama 2 and Mistral, which align with our experimental setup.
> > > - GEM (full name: "Entropic Distribution Matching in Supervised Fine-tuning of LLMs: Less Overfitting and Better Diversity"): Their main experimental discussion was conducted on Llama-7B.
> > > - DPO: Evaluations were performed on GPT-J and Pythia, also selecting two representative models of that time.
> > >
> > > Considering the experimental validation from these related works, we believe that our experimental results on Llama 3 and Mistral models may be sufficient to demonstrate the effectiveness of our method.
> > >
> > > ## About performance decrease on a few benchmarks
> > > Regarding why performance decreases on some benchmarks, this has been discussed in previous work, such as SimPO, so we did not provide an extensive explanation. The main fluctuations stem from AI evaluation itself, and similar decreases can be observed across many benchmarks in SimPO as well.
> > >
> > > Despite the random fluctuations, we want to claim that our method shows better performance on average. Meanwhile, performance improvements are very significant on most datasets. We hope this addresses your concerns.
> > >
> > > ## Regarding the selection of preference learning variants
> > >
> > > To validate the effectiveness of our theory, we chose two of the most representative methods for demonstration. DPO represents the most commonly used method, while SimPO represents the best-performing method as of our paper's submission deadline. We achieved excellent results in both of these experiments.
> > >
> > > If you still expect to see results from other variants, we will try to follow up further in subsequent discussions. Due to the time constraints of the discussion period, we regrettably cannot guarantee it.
> > >
> > > Finally, we thank you for your active participation in the discussion. And we value your feedback greatly. Our work combines theory and practice, with approximately half of the content dedicated to describing our theoretical contributions. We sincerely invite you to consider our contributions comprehensively, taking into account both our theoretical and experimental results.

---

> > > > ### Comment · Reviewer_SPrv · 2025-08-07
> > > >
> > > > Thank you for your prompt and detailed reply.
> > > >
> > > > First of all, I would like to express my disappointment that my initial W5 comment — which indeed contained a copy-paste typo — led to the perception that all of my review might be AI-generated, and consequently resulted in an incomplete rebuttal in the first round. While I acknowledge my mistake in W5, I would hope that the rest of the weaknesses and questions in my review could have been treated on their own merits.
> > > >
> > > > Second, while I understand that a significant portion of your paper is dedicated to theoretical contributions, my comments focus primarily on the experimental validation because I believe that strong theoretical insights require equally thorough empirical evidence to be meaningful in practice. From my perspective, the current experimental evaluation does not yet meet that bar, which raises concerns about the practical significance of the proposed theory.
> > > >
> > > > Regarding the learning rate: I appreciate your additional experiments with a smaller learning rate (1e-6). However, your current justification that "smaller learning rates are generally better" still lacks a clear rationale or principled guideline for choosing what constitutes "small". Without this, the conclusion feels somewhat arbitrary, and my concern remains unresolved.
> > > >
> > > > On model selection: While I understand that adding more models can be time-consuming, I still believe that including at least one additional model—either from a different size or a different family—is necessary to enhance the generality of your empirical findings. As seen in other recent works (e.g., KTO), such configurations have become a standard practice.
> > > >
> > > > As for performance degradation on certain benchmarks: If specific benchmarks are known to suffer from instability due to AI evaluation fluctuations, it raises the question of whether they should be included in the first place, especially if no meaningful conclusion can be drawn from them. I would encourage a more thoughtful reflection on the choice of evaluation datasets in future iterations.
> > > >
> > > > In summary, I appreciate the effort to bridge theory and practice in this work. However, I believe that theoretical contributions must be accompanied by robust empirical validation to fully demonstrate their value.

---

> > > > > ### Author Response · Authors · 2025-08-07
> > > > >
> > > > > Thank you for your deep attention to our work. At the same time, we would like to apologize again for the previous misunderstanding.
> > > > >
> > > > > ## About Learning Rate
> > > > >
> > > > > Your concern about the scale of what constitutes 'small' is a very interesting question. Just like the analogy with adding KL regularization terms—although KL optimization is generally considered effective in RL, practice still requires some hyperparameter search. We believe that discussing the learning rate range for small learning rates might be more appropriate as a separate future work. It may be better to consider that we provide a theoretically guaranteed search direction, with the theoretical guarantee being KL properties, and this search direction is feasible.
> > > > >
> > > > > ## About Model Selection
> > > > >
> > > > > We fully understand your concern about model selection, and as you mentioned, supplementing with new model selections is very time-consuming. We will do our best to supplement within the remaining time, and we hope you can reconsider the sufficiency of the existing experiments comprehensively. Thank you again for your understanding.
> > > > >
> > > > > ## About Benchmark Selection
> > > > >
> > > > > Your criticism about benchmarks is very insightful. Our benchmark selection is primarily based on considerations of fair comparison, choosing to maintain consistency with other related work. We do not deny that these benchmarks have their issues, but from the perspective of community fair comparison and comprehensive consideration of existing evaluation schemes, the current flawed evaluation schemes may already be the better options. We will also consider developing benchmark research as a separate future work. Thank you for your correction.
> > > > >
> > > > > Finally, we sincerely apologize for the misunderstanding in our previous communication, which caused you distress and we deeply regret that. We greatly appreciate your feedback on our work and the time you spent participating in the discussion. In our previous replies, we tried to compare our work with related work. We sincerely hope you can reconsider your evaluation of our work comprehensively. We will try to supplement with new model families according to your suggestions, but we cannot guarantee the timeline. We hope this will not affect your comprehensive re-evaluation of our work. Thank you!

---

> > > > > > ### Author Response · Authors · 2025-08-08
> > > > > > **Additional Experimental Results**
> > > > > >
> > > > > > We sincerely apologize for bothering you again! We greatly respect your concern about whether the model selection is adequate, and as requested, we have rushed out some new experimental results, which are as follows:
> > > > > >
> > > > > > ## New Setting
> > > > > >
> > > > > > Following your considerations regarding model family and model size, we selected Qwen2.5-3B as the base model. We conducted SFT and DPO training on both the Ultrachat-200K and Ultra-feedback datasets. For the baseline, the SFT learning rate was set to 2e-5, while the DPO learning rate was set to 5e-7. The smaller learning rate remained at 5e-6, with all other training configurations consistent with previous settings. During decoding, we used temperature = 0.7 and top_p = 0.95. The evaluation results are as follows:
> > > > > >
> > > > > > ## Results
> > > > > >
> > > > > > | Model | Alpaca Eval 2 | | Arena-Hard | MT-bench | |
> > > > > > |:------|:-------------:|:-------------:|:-----------:|:---------:|:---------:|
> > > > > > | | LC-win | Win | Win | GPT-4 | GPT-4-turbo |
> > > > > > | SFT | 5.06 | 2.86 | 3.1 | 6.1 | 5.9 |
> > > > > > | DPO | 8.82 | 6.78 | 9.9 | **7.0** | **6.9** |
> > > > > > | **SFT(small lr)** | **5.7** | **3.32** | **3.7** | 5.9 | 5.8 |
> > > > > > | **DPO(small lr)** | **10.79** | **8.73** | **11.6** | **7.0** | **6.9** |
> > > > > >
> > > > > > Based on the new results, searching towards smaller learning rates on Qwen 2.5-3B also proves to be a good search direction
> > > > > >
> > > > > > We are delighted to have rushed out our new results before the discussion concludes. We fully respect your attention to our work and hope these results can address your concerns to some extent. If you would be willing to adjust your score accordingly, we would be very grateful.

---

> > > > > > > ### Comment · Reviewer_SPrv · 2025-08-09
> > > > > > >
> > > > > > > Thank you for your additional efforts. I find the theoretical framework of this paper interesting. However, I still have doubts about the practical value of the claim that “a smaller learning rate generally leads to better performance.” Although the authors added a new model, I do not see its results presented in Table 3. Overall, I still consider the practical applicability of this work to be limited, and the current experiments are insufficient to fully support its theoretical framework (align with reviewer 4L4H). Therefore, I will keep my score unchanged.

---

> > > > > > > > ### Author Response · Authors · 2025-08-09
> > > > > > > >
> > > > > > > > Thank you for actively participating in the discussion. The SFT training for the model in Table 3 has been relatively slow; due to time constraints, we have not yet obtained its results for inclusion in Table 3.
> > > > > > > >
> > > > > > > > ## Regarding how small the learning rate should be
> > > > > > > > We would like to emphasize that even for the KL regularization term, it is difficult to determine the optimal beta coefficient without a search. This does not diminish the effectiveness of KL itself. A smaller learning rate is simply a theoretically motivated search direction to alleviate KL pressure, and it happens to align with practice.
> > > > > > > >
> > > > > > > > ## On the consistency between theory and experiments
> > > > > > > > Beyond recommending a small learning rate, our theoretical framework contributes in several other ways, including deriving training objectives for alternative frameworks, extending properties of QV, and offering an explanation for why SFT remains necessary. The agreement between theory and practice from multiple angles collectively substantiates the value of our work.
> > > > > > > >
> > > > > > > > Focusing only on the small learning rate, we have already detailed that our results are fully consistent with related prior work. And we have added new experimental results as requested. We hope that, taken together, these points make the effectiveness of our work more convincing.
> > > > > > > >
> > > > > > > > Overall, we are deeply grateful for your active engagement in the discussion, and we sincerely thank you again for your time.

---

> ### Author Response · Authors · 2025-08-06
> **Have We Answered Your Questions?**
>
> We apologize for the interruption. We would like to know if our reply has addressed your questions or if you need any further clarification from us.

---

### Official Review · Reviewer_dTok · 2025-07-01

**Clarity:** 3
**Significance:** 2
**Originality:** 3
**Rating:** 4
**Confidence:** 4

**Summary:**

The paper attempts to connect SFT and preference learning through the idea of the implicit reward and describes SFT as distribution matching using the TV distance without a KL term. Through this, they describe SFT as searching the optimal policy-reward subspace and suggest using smaller learning rates for SFT. They also provide alternatives to standard SFT derived from different f-divergences. They claim that the logits after SFT correspond to state quality. Experiments are performed to verify these claims.

**Questions:**

- Can you provide some justification for Assumption 1? as well as for why you expect the cross-entropy between the policy and the reference to be small? Output probabilities do change significantly over the course of SFT especially if introducing a chat format.
- Can you provide some clarification on eq. 4? What is $s_t$? and what is meant by $V^*$ is the value function of the optimal policy?
- Could you provide experimental results for SFT with the KL term?

Justification on the assumptions and steps taken are the current main concern and better explanations or experiments demonstrating that these steps are reasonable would increase the score.

**Ethical Concerns:**

["NO or VERY MINOR ethics concerns only"]

**Final Justification:**

Results are mostly valid with new insights of SFT

**Limitations:**

The assumptions necessary may be a limitation that needs to be addressed.

**Paper Formatting Concerns:**

No formatting concerns

**Quality:**

2

**Strengths And Weaknesses:**

Strengths:
- Background on distribution matching and the problem setting was clearly presented with definitions and explanations.
- The paper provides a novel analysis that provides an understanding of the connection between SFT and DPO which is a common part of the language model training pipeline.
- The paper overall clearly states results and assumptions with empirical verification of results.

Weaknesses:
- There are some steps that need further justification. Mainly Assumption 1 which makes a rather strong assumption about the value function and the claim about the KL term in the derived SFT being treated as a constant with little effect. It seems unclear how true or practical these claims are and seem to be an important part of being able to draw the connection using the implicit reward. This is a large weakness especially since the claim about SFT searching the optimal policy-reward completely depends on this assumption and following results build on this. Justification for replacing entropy with KL and why the cross-entropy term is small is currently weak.
- Using the identity function as the conjugate function for TV only holds over a small range so if pi starts to deviate more from the initial model, this connection is no longer valid and also further suggests that the KL term in SFT could have a non-trivial effect.
- It seems in the comparisons with other objectives from f-divergences, SFT is used without the KL term which seems like an unfair comparison. Also SFT with the KL term is not used in any of the experiments.

---

> ### Author Rebuttal · Authors · 2025-07-29
>
> Thank you for taking the time to read our work. Your judgment about the importance of KL is very accurate. There may be some aspects we haven't expressed clearly, so we would like to restate our method here:
> - The post-training process typically includes multiple learning methods such as SFT and RL, and their theoretical connections are underexplored. To deeply understand what the SFT process is actually doing, we return to the maximum entropy imitation learning optimization objective and make corrections that align with language modeling, adding a cross-entropy term to the entropy term. The goal is to keep the model from deviating from the language distribution while maintaining explorability.
> - We follow the derivation approach of non-adversarial imitation learning and obtain a series of imitation learning objectives. When the f-divergence is chosen as total variance, it exactly equals the commonly used SFT objective function. We also obtained some new insights about the SFT process, which we list as follows
> - The above derivation process proves that SFT is actually also learning implicit reward. According to Equation 2, SFT is equivalent to searching for the most suitable reward in the optimal reward-policy space based on the given expert demonstrations. At the same time, DPO also proves that in reference learning, we can learn implicit rewards to directly fit preference signals. So implicit reward can naturally serve as a unified perspective to connect with DPO.
> - We found that when taking total variance, the original KL derivative term -log π_ref is a zero-order term in stochastic gradient descent and does not contribute to the gradient. In RL setting, the role of KL is mainly to constrain the magnitude of single-step updates. We constrain the step size of single-step optimization in the same way by reducing the learning rate.
> - Since the derivation process has the same mathematical structure as DPO, we simultaneously extend the connection between LLM logits and Q-function from the DPO process to the SFT context. We found that the logits of LLM during the SFT process can also serve as a Q function. They represent the expected future return starting from the current state and can serve as a judgment of the quality of the current state-action pair.
>
> ## About the cross-entropy term in KL will be small
>
> You proposed a thoughtful question! In fact, this value is relative. When LLM pre-training converges, the cross-entropy loss typically ranges from 1.x to 2.x(https://arxiv.org/pdf/2407.21783). For the original maximum entropy, taking Qwen(https://arxiv.org/abs/2309.16609) as an example, the entire vocabulary size is 128,256, so the maximum entropy at uniform distribution is log(1/128,256) = 5.11, which is significantly larger than the cross-entropy loss. At this point, it completely loses the characteristics of language distribution. Therefore, although the SFT stage changes output probabilities, as long as the output is natural language, the cross-entropy loss constraint is reasonable.
> ## What is $V^*$
>
> For any given reward, there must be a corresponding optimal policy. The expected future return that the current policy will obtain in a certain state is defined as $V^*$.
>
> ## The KL deficiency problem in SFT
>
> Your judgment about KL deficiency in SFT is absolutely correct! This is also one of the core contributions of our work. We found that the KL term, which originally plays a huge role in the imitation learning objective, becomes ineffective due to the corresponding term under total variance, i.e., -log π_ref in Equation 5, being a zero-order term. Therefore, we made some corrections to common SFT tricks, including small learning rate correction and other imitation loss corrections. We found that these effects all show significant improvements.
>
> Finally, thank you again for your insights, and we look forward to further discussion with you!

---

> > ### Comment · Reviewer_dTok · 2025-08-03
> >
> > Thank you for the clarifications about KL being the zeroth-order term in the SFT derivation. My concerns around missing the KL term in experiments are addressed. However, my concerns around Assumption 1 and the validity of the identity being the conjugate function for TV are unaddressed. Furthermore, it remains unclear the the cross-entropy term is small compared to the entropy term, especially since if $\pi_\text{ref} = \pi$, they are the same. From this starting point, it seems that the cross-entropy and entropy terms would change at comparable rates and neglecting one of the terms would likely cause significant changes. I will be keeping my score.

---

> > > ### Author Response · Authors · 2025-08-04
> > > **Sorry for the Confusion. Here are some clarification and further discussion**
> > >
> > > We're pleased that we've successfully resolved your concerns about the missing SFT issue. We also feel sorry that we didn't fully address your confusion last time. We hope the following content can address your remaining questions.
> > >
> > > ## About KL term
> > > Your understanding that the inclusion of the cross-entropy term is non-trivial is completely correct. In our previous discussion, we did not properly understand your point, and we would like to apologize and provide a proper explanation of this matter.
> > >
> > > - In traditional imitation learning, to ensure exploration, an additional entropy term is typically added as a regularization term. This is what's known as maximum entropy imitation learning, with one of the most famous works being Soft Actor-Critic.
> > >
> > > - Entropy regularization applies an unbiased constraint across the entire action space. For LLMs, the action definition in the MDP process typically covers the entire vocabulary range. An unmodified entropy constraint would cause the model to deviate from the language distribution.
> > >
> > > - From an objective perspective, we want to ensure LLM's exploratory behavior, but it should be explored within a reasonable language distribution. Therefore, we add a cross-entropy term as an additional optimization regularizer.
> > >
> > >  As you correctly pointed out, this is a non-trivial modification to the optimization objective. The goal is to leverage the property that large language models have seen extensive natural language distributions during their pre-training phase, ensuring that exploration occurs within a space that conforms to natural language distributions.
> > >
> > > Similar concepts are also mentioned in the work 'The Entropy Mechanism of Reinforcement Learning for Reasoning Language Models.' We hope the above explanation resolves your confusion.
> > >
> > > ## About assumption 1
> > > Assumption 1 is primarily an assumption about the initial reward function.
> > >
> > > When deriving to Theorem 1, we can see that the optimization process is divided into two steps: the inner optimization finds the optimal policy for any given reward function, followed by the process of optimizing the reward function. In this derivation process, the starting point of optimization, which is the reward in the first step, is arbitrary. This allows us to add Assumption 1, namely that the initial reward function is numerically equal to V(s). This assumption only specifies what kind of reward function the policy model is optimal under during the first step of optimization.
> > >
> > > ## About the relationship between assumption and derivation process
> > >
> > > There are some aspects regarding the relationship between KL and the overall theoretical derivation that we haven't clearly articulated:
> > >
> > > - From the first optimization objective at line 118 to the form of Theorem 1 at line 138, the derivation does not depend on the properties of KL. This part mainly transforms the original optimization objective through the properties of conjugate functions and saddle points. The entropy term is still a concave function. The derivation in Appendix B.1 can proceed normally by simply replacing all KL parts with entropy terms.
> > >
> > > - The main role of KL is that it lead to the closed-form solution form of the implicit reward, which is particularly important in the derivation from Eq. 2 to Eq. 5. We have also mentioned the rationality of KL earlier—the goal is to make a non-trivial modification to the commonly used maximum entropy form that conforms to the language distribution.
> > >
> > > ## About Total Variation and its corresponding conjugate function
> > >
> > > We apologize for the misunderstanding caused by the simplified content. Although there is a one-to-one correspondence between the conjugate function f* and the generator function f, logically it's not that we first have the identity as the conjugate function. Rather, we first have the Total Variation variant of f-divergence, whose corresponding conjugate function happens to be the identity function. Under Total Variation, its corresponding generator function and f-divergence form are as follows.
> > >
> > > The f-divergence is $\frac{1}{2} \int |p(x)- q(x)|dx$. The generator function is $f(u) = \frac{1}{2} |u -1|$ and corresponding $f^*$ is identity function
> > >
> > > For the simplicity of the table, we simplified this content and referenced the aforementioned work f-GAN at lines 160, 558, and 572 as supplementary material.
> > >
> > > This is also mentioned in our references, the aforementioned work f-GAN. In the derivation of f-GAN, Total Variation is also an optional GAN form, and when Total Variation is chosen, the optimization objective under derivation happens to be the SFT objective function. Total Variation is a valid f-divergence option that leads to a common SFT form. But your questioning of potential issues with Total Variation is very reasonable. And we also claim that the current SFT has problems.
> > >
> > > We apologize for not fully resolving your confusion earlier. Thank you for your time, and we genuinely look forward to further discussion.

---

> > > > ### Comment · Reviewer_dTok · 2025-08-04
> > > >
> > > > Thank you for the response. My concern on Assumption 1 has been addressed.
> > > >
> > > > However, the idea that an unmodified entropy constraint would cause the model to deviate from the language distribution does not seem true, especially given that beta is a parameter that can be chosen appropriately. I think this justification should either be removed or a stronger argument should be used.
> > > >
> > > > Regarding the TV distance, my concern is that the conjugate function for the TV is the identity only over a small bounded region and infinite elsewhere. While this does not matter for the optimal policy, when choosing an objective based on this, it seems likely that the reward will go outside of this region, making the connection between SFT and the TV-based objective less direct.
> > > >
> > > > I think further clarification and justification are needed for rigorous results that are fully correct, but after further discussion, it seems the insights mostly hold under reasonable conditions, so I will increase my score.

---

> > > > > ### Author Response · Authors · 2025-08-04
> > > > >
> > > > > We are glad our reply resolved your confusion. Thank you again for your support of our work！

---

### Official Review · Reviewer_GAjJ · 2025-07-03

**Clarity:** 3
**Significance:** 3
**Originality:** 4
**Rating:** 5
**Confidence:** 4

**Summary:**

Can we understand supervised fine-tuning (SFT) and direct preference optimization (DPO) as part of a unified reward-based framework? The paper present argues that SFT is in fact optimizing an implicit reward function, just like DPO does explicitly. This reframes both SFT and DPO as reward-based methods under a shared mathematical umbrella. The authors show that SFT corresponds to inverse reinforcement learning (IRL) by matching expert distributions. They point out a limitation of SFT, where the KL term in distribution matching becomes constant during training—making SFT unable to constrain model updates effectively. To address this, they advocate using alternative divergences (e.g., Pearson χ²) to recover stable reward shaping and show that this leads to better downstream DPO performance. They also generalize prior work connecting LLM logits and Q-functions in preference learning, extending it to SFT as well. Experiments with LLaMA-3 and Mistral on UltraChat/UltraFeedback validate their theory.

**Questions:**

# Questions

1. Could the unified framework potentially support more complex multi-turn feedback protocols beyond binary preferences (e.g., fine-grained ratings, trajectory-level reward models)?
2. Is there a formal characterization of when the logits-to-Q-function approximation in SFT breaks down? Under what training regimes or tasks does this interpretation become unreliable?

**Ethical Concerns:**

["NO or VERY MINOR ethics concerns only"]

**Final Justification:**

After carefully reviewing the papers rebuttal and other conversations on this thread I maintain my score.

**Limitations:**

Yes

**Quality:**

4

**Strengths And Weaknesses:**

# Strengths

* The paper theoretically unifies two central components of LLM alignment — SFT and DPO — under a shared framework based on implicit rewards and distribution matching.
* It clearly identifies a flaw in conventional SFT training: the constant KL term fails to regulate model updates and can harm downstream performance. This is a non-trivial insight with actionable implications.
* It goes beyond theory and proposes two concrete techniques backed by experiments: (1) reducing the SFT learning rate, and (2) switching to divergences (χ², Squared Hellinger) that preserve gradient signal.
* Empirical results validate theory across multiple models (Mistral, LLaMA-3) and evaluation sets (AlpacaEval, Arena-Hard, MT-Bench), showing large relative win rate improvement post-DPO.
* The extension of the Q-value/logit interpretation to SFT broadens its usefulness for reward estimation and could inspire further work on early-stage reward shaping diagnostics.

# Weaknesses

* The experimental validation is focused on a narrow domain: instruction tuning with UltraChat and UltraFeedback. It's unclear if the same effects hold in tasks with fundamentally different reward structures (e.g., code generation, tool use, multi-turn dialog).
* The Q-function interpretation of logits in SFT is supported mostly by correlation experiments rather than rigorous causal or ablation-based testing. No direct test of how useful this proxy is in practice.

---

> ### Author Rebuttal · Authors · 2025-07-29
>
> We are very grateful that you are willing to spend time carefully reading our work and for your high recognition of this research.
>
> ## About Different Reward Structure
>
> Our work mainly provides theoretical relevance for understanding the SFT process from the perspective of implicit reward. Thus, alignment algorithms that use implicit reward can easily establish connections with our theoretical framework. For different reward structures like complex multi-turn feedback, there are already some works based on implicit reward, such as https://arxiv.org/pdf/2406.14868, that can establish connections with our work via implicit reward. At the same time, because in our imitation learning process, we target the finest-grained reward function, i.e., token-level reward, trajectory reward, or step-level reward can all be obtained through additive aggregation from token-level reward, allowing for very natural theoretical extensions.
>
> ## About Logits-to-Q
>
> Your interest in the logits-to-Q extension is very insightful. As we mentioned in L187, the logits-to-Q properties in the DPO process have already been pointed out in previous work. Our work derived a consistent mathematical structure in the SFT process and extended the theory. There are works(https://arxiv.org/abs/2502.01456, https://arxiv.org/abs/2404.18922) that have already used the logits-to-Q properties in the DPO process to guide the learning process and achieve good results.
> Practically, we found that in the experiments conducted in Section 4.4, the logits of LLMs show greater differences when they have consistent judgments(Avg std. 5.81) about quality compared to when they have opposite judgments (Avg std. 2.76). Since the statistical data shows that the quality judgments of the two models are positively correlated in the experiments, we suggest treating the conclusion of non-validity as random fluctuations caused by tokens having insignificant contributions to the final reward. We hope this can make the logits-to-Q question more convincing.
>
> Thank you again for taking the time! We are very grateful for your recognition of our work!

---

> ### Comment · Reviewer_GAjJ · 2025-08-07
>
> Thank you for the response. I appreciate the argument that token-level rewards can extend naturally to multi-turn or fine-grained feedback settings, as well as empirical hints you provided on the logits-to-Q property. It would be reassuring if the narrowness of the task domain in the experiments were addressed as well.

---

> > ### Author Response · Authors · 2025-08-08
> >
> > Thank you for recognizing and supporting our work! We're pleased to have solved your problem and will continue expanding into more domains in the future.

---

### Official Review · Reviewer_4L4H · 2025-07-03

**Clarity:** 3
**Significance:** 2
**Originality:** 3
**Rating:** 3
**Confidence:** 3

**Summary:**

This paper reframes the relationship between SFT and DPO, arguing they both exist in an "optimal policy-reward subspace" where an implicit reward is being learned. The core of the work is deriving the SFT objective from an f-divergence minimization problem, showing that SFT corresponds to using the total variation distance. This viewpoint highlights that SFT lacks the explicit KL-regularization present in DPO and other RL algorithms. The authors propose that this missing regularization can be compensated for by using a small learning rate or by adopting alternative f-divergence objectives for SFT. Experiments show these modifications to the SFT stage improve the final performance of a model after it undergoes DPO.

**Questions:**

1. The paper frames DPO as navigating along a "chosen-rejected" direction and SFT as moving along an "average direction" (Figure 1). Could you formalize what this "average direction" means in the context of your implicit reward framework? Is it the gradient of the expected reward over the expert demonstration distribution?
2. How does your framework relate to other methods that modify the SFT objective, such as ORPO, which combines the SFT and preference-learning objectives into a single loss function? Does your theory offer any insight into why such approaches might work?
3. You show that better regularization during SFT leads to a better starting point for DPO. Does this imply that the reference model used in DPO should ideally be the final, regularized SFT model rather than the original pre-trained base model?

**Ethical Concerns:**

["NO or VERY MINOR ethics concerns only"]

**Final Justification:**

While the paper provides fresh theoretical perspectives on SFT and DPO, the practical validation is not thorough enough (e.g., on old and small base models), and the improvement is relatively marginal. Therefore, I am not confident in its significance.

**Limitations:**

yes

**Quality:**

3

**Strengths And Weaknesses:**

Strengths:
- The paper does a good job of building a conceptual and mathematical bridge between SFT and DPO, which are often treated as separate, sequential steps. The "implicit reward" framing is interesting.
- The analysis pinpoints the theoretical weakness in the standard SFT objective of lacking a KL constraint, which explains some of the known drawbacks of SFT.
- The improvements from reducing the learning rate are clearly demonstrated. The finding that alternative f-divergences can also improve post-DPO performance (Table 3) is a compelling piece of evidence for the overall theory.
- The paper is well-written and well-structured. Figure 1 provides an excellent schematic for the core idea, and the mathematical arguments are sound.

Weaknesses:
- The extension of the logits-as-Q-function theory to SFT (Theorem 2) relies on Assumption 2. This assumption is quite strong and is not directly validated. A more thorough discussion of when this assumption might hold or fail would strengthen the claim.
- A fascinating result in Table 3 is that the Pearson and SH-SFT models, which perform worse than the standard SFT model on their own, produce better post-DPO models. This strongly supports the paper's thesis about regularization but is a non-obvious finding. The paper could benefit from a slightly more detailed discussion of this implication: that the best SFT model (by imitation metrics) may not be the best foundation for preference tuning.
- The main practical takeaway is to use a smaller learning rate for SFT. While the paper provides a new theoretical justification for this, using smaller learning rates for more stable fine-tuning is a common heuristic. The novelty is therefore more in the "why" than the "what," and the practical impact could be seen as an incremental improvement on existing best practices.
- The framework provides a unified perspective but does not result in a unified algorithm. The SFT->DPO pipeline remains sequential. The paper shows that doing SFT "better" helps DPO, but the two objectives are still optimized separately. The failed multi-objective experiments underscore that a true unification is still an open problem.

---

> ### Author Rebuttal · Authors · 2025-07-29
>
> Thank you for your patient reading and your very insightful questions.
> ## About logits-to-q
>
> Your question about logits-to-q is very perceptive. There are three points we would like to discuss with you:
> 1. As we mentioned in L187, the relationship between logits and the q function in the DPO process has also been discussed in previous works(https://arxiv.org/abs/2404.18922, https://arxiv.org/pdf/2404.12358). We derived a consistent mathematical structure based on the SFT optimization objective and extended the logits-to-q conclusion from the DPO process to the SFT process. Assumption 2 is similar to that made in previous work.
> 2. Beyond theoretical proof, we provide indirect experimental evidence in Section 4.4. Since Q-V functions represent expected returns and their numerical values indicate the quality of state-action pairs, we showed that different models share some common judgments in their quality assessments. This indirectly proves the correctness of logits-to-q.
> 3. After thoroughly investigating the validity of logits-to-q, we find that the magnitude of logits changes is more pronounced (Avg std. 5.81) when two models share similar judgments, compared to when they have opposite assessments of state quality (Avg std. 2.76). This also supports the validity of Assumption 2.
> We hope these explanations can make the logits-to-q conclusion and corresponding assumption 2 more convincing.
>
> ## A better SFT starting point for DPO may not show better performance on metrics
>
> Thank you for your interest in the phenomenon we proposed. To the best of our knowledge, the commonly used methods for selecting starting points for DPO are greedy. The conclusion we proposed in L268 may serve as a contribution to correcting misconceptions when selecting the starting point for DPO. Our explanation for this is that these SFT checkpoints may not be the best ones for performing a specific task, but they maintain better diversity in responses, i.e., higher entropy. This makes these SFT checkpoints better starting points
>
> ## More new insights beyond the small learning rate provided by theoretical analysis.
>
> Your insight about empirically using small learning rates to stabilize fine-tuning during post-training is very perceptive. Our theoretical analysis obtains the same methods from a KL perspective, which can serve as a theoretical explanation. Beyond this, we also proposed a series of imitation learning objectives that maintain KL properties in Section 4.3, analyzed the implicit reward properties during SFT in Section 4.4, and attempted to answer the role of SFT in post-training in Section 4.5. We hope these new insights from theoretical analysis can also excite you!
>
> ## About the unified algorithm combining SFT and DPO
> The failure to combine the SFT and DPO processes into one process is regrettable. However, we have made some progress toward this goal in this work, as we have obtained some theoretical guarantees. There are many possible reasons for the failure of multi-objective optimization, and we believe the most likely cause is data quality. We leave this to future work and hope our progress can inspire subsequent insights.
>
> ## About the average direction in Figure 1
> You are absolutely correct. Here, "average direction" means that, assuming there are four data points in a batch, as shown in Figure 1 of our paper, these four data points contribute equally to the estimation of the optimization direction, that is, to the estimation of the reward gradient.
>
> ## Some more theoretical proof of ORPO
> Your questions about other methods are interesting! We provide some extended theoretical analysis here about ORPO. The original optimization objective of ORPO is written as:
> $$L = L_\text{SFT} + \lambda L_{OR}$$
> The first part is the classic SFT loss, which can be transformed into a reward learning process through the same approach. The second part can be expanded and written as:
> $$L_\text{OR} = \log \sigma(\log \frac{\frac{P(y_w|x)}{1-P(y_w|x)}}{\frac{P(y_l|x)}{1-P(y_l|x)}} )$$
> Consider the term inside the $\log \sigma$
> $$\log \frac{\frac{P(y_w|x)}{1-P(y_w|x)}}{\frac{P(y_l|x)}{1-P(y_l|x)}} $$
> $$= \log \frac{P(y_w|x)}{1-P(y_w|x)} - \log \frac{P(y_l|x)}{1-P(y_l|x)} $$
> $$= \log \frac{P(y_w|x)}{P_\text{ref}(y_w|x)} - \log \frac{P(y_w|x)}{P_\text{ref}(y_l|x)} + \log \frac{1-P(y_l|x)}{1-P(y_w|x)} + \log \frac{P_\text{ref}(y_w|x)}{P_\text{ref}(y_l|x)} $$
> $$= \log \frac{P(y_w|x)}{P_\text{ref}(y_w|x)} - \log \frac{P(y_w|x)}{P_\text{ref}(y_l|x)} + \log \frac{\Sigma_yP(y|x) - P(y_l|x)}{\Sigma_y P(y|x) - P(y_w|x)}+ \log \frac{P_\text{ref}(y_w|x)}{P_\text{ref}(y_l|x)} $$
> $$= \log \frac{P(y_w|x)}{P_\text{ref}(y_w|x)} - \log \frac{P(y_w|x)}{P_\text{ref}(y_l|x)} + \log \frac{\Sigma_y\frac{P_\text{ref}(y|x)}{P_\text{ref}(y_l|x)}\frac{P(y|x)}{P_\text{ref}(y|x)} - \frac{P(y_l|x)}{P_\text{ref}(y_l|x)}}{\Sigma_y \frac{P_\text{ref}(y|x)}{P_\text{ref}(y_w|x)}\frac{P(y|x)}{P_\text{ref}(y|x)} - \frac{P(y_w|x)}{P_\text{ref}(y_w|x)}} $$
> $$=r_w - r_l + \log \frac{\Sigma_y\frac{P_\text{ref}(y|x)}{P_\text{ref}(y_l|x)}\exp(r(x, y)) - \exp(r(x, y_l))}{\Sigma_y \frac{P_\text{ref}(y|x)}{P_\text{ref}(y_w|x)}\exp(r(x, y)) - \exp(r(x, y_w))} $$
>
> The $r_w - r_l$ is the classic BT model, aiming to widen the difference in reward between the chosen sample and the rejected sample. The latter part takes the total amount of reward into account. After dividing both the numerator and denominator by $\Sigma_y P_\text{ref}(y|x) \exp(r(x, y))$, we obtain:
> $$ r_w - r_l + \log \frac{\frac{1}{P_\text{ref}(y_l|x)} - \frac{\exp(r(x, y_l))}{\Sigma_y P_\text{ref}(y|x) \exp(r(x, y))}}{\frac{1}{P_\text{ref}(y_w|x)} - \frac{\exp(r(x, y_l))}{\Sigma_y P_\text{ref}(y|x) \exp(r(x, y))}} $$
> $$ =r_w -r_l + lop \frac{c_1 - \text{ratio of } r_l}{c_2 - \text{ratio of } r_w} $$
> This can be viewed as a correction term that represents, beyond increasing the reward difference between win and loss samples, further consideration of the total reward allocation for other possible responses. It considers the relationship between the current sample's reward allocation and the total reward allocation, requiring that the win portion accounts for a higher proportion of the total reward allocation than the loss portion, thereby further constraining the reward learning process.
>
> ## Choosing KL-SFT checkpoint as the reference model in DPO is better
>
> You reach an absolutely correct conclusion about the reference model selection during the DPO process. To the best of our knowledge, current DPO implementations have already begun using the model after the SFT process as the reference model instead of the base model. In our work, we further showed that the KL-constrained SFT checkpoint is a better choice for the starting point of DPO. This phenomenon maintains high consistency with our theoretical analysis.
>
> We hope the above response finds you well. Thank you again for your time for our work.

---

> > ### Comment · Reviewer_4L4H · 2025-08-08
> >
> > Thank you for the detailed response. While the paper provides fresh theoretical perspectives on SFT and DPO, the practical validation is not thorough enough (e.g., on old and small base models), and the improvement is relatively marginal. Therefore, I am not confident in its significance and will retain a borderline rating.

---

> > > ### Author Response · Authors · 2025-08-08
> > >
> > > Thank you for your response and support of our work! We hope our previous replies have addressed most of your concerns. Below are some supplementary experimental results, which we hope will address your concerns about the practical effectiveness.
> > >
> > > ## About Experiment
> > >
> > > Our current experimental setup is largely consistent with other related work. The Llama3 series models were released around this time last year. To ensure fair comparison by maintaining consistency with other work, we selected the Llama3 and Mistral model families, while also ensuring consistency in model size.
> > >
> > > To further validate the effectiveness of our method, we have just obtained new results on the new model Qwen 2.5-3B as supplementary experimental results to further demonstrate the effectiveness of our approach.
> > >
> > > ### New Setting
> > > We selected Qwen2.5-3B as the base model and conducted SFT and DPO training on both the Ultrachat-200K and Ultra-feedback datasets. The SFT learning rate was set to 2e-5, while the DPO learning rate was set to 5e-7. The smaller learning rate was chosen as 5e-6, with all other training configurations remaining consistent with previous settings. During decoding, we used temperature = 0.7 and top_p = 0.95. The model evaluation results are as follows:
> > >
> > > | Model | Alpaca Eval 2 | | Arena-Hard | MT-bench | |
> > > |:------|:-------------:|:-------------:|:-----------:|:---------:|:---------:|
> > > | | LC-win | Win | Win | GPT-4 | GPT-4-turbo |
> > > | SFT | 5.06 | 2.86 | 3.1 | 6.1 | 5.9 |
> > > | DPO | 8.82 | 6.78 | 9.9 | **7.0** | **6.9** |
> > > | **SFT(small lr)** | **5.7** | **3.32** | **3.7** | 5.9 | 5.8 |
> > > | **DPO(small lr)** | **10.79** | **8.73** | **11.6** | **7.0** | **6.9** |
> > >
> > > Through the results, we can further observe that our method achieves results that meet expectations across different model sizes and families. We hope our new results will make our work more convincing to you.

---

> ### Author Response · Authors · 2025-08-06
> **Have We Answered Your Questions?**
>
> We apologize for the interruption. We would like to know if our reply has addressed your questions or if you need any further clarification from us.

---

### Note · Authors · 2025-08-13

## Our Main Contributions
- We contribute a unified theoretical perspective from the implicit reward angle and provide rigorous mathematical proofs
- We identify KL deficiency in existing SFT loss and propose corrections via learning rate reduction or loss modification
- We theoretically extend the logits-to-Q property for LLMs
- We experimentally demonstrate:
  - Small learning rates and loss modifications are effective
  - Indirect proof of the logits-to-Q property
  - Why SFT remains necessary
## Resolved Concerns
### Assumption 1 was misinterpreted with overly strong V assumptions
- We clarify Assumption 1 relates to initial rewards, not V requirements
### Assumption 2 appears overly strong for logits-to-Q
- We explain that Section 4.4 indirectly proves it holds in most cases
- We supplement numerical characteristics: V shows greater variation trends when Assumption 2 holds
- We reference pre-extension conclusions for justification
### Whether select TV is correct in f-divergence
- We cite f-GAN work
- Exp showing TV is valid in most cases
### Whether Cross-entropy modification is necessary
- We explain LLM exploration should occur in natural language space, CE constraints hold in most cases
### Framework extension to complex reward structure
- We demonstrate the simplicity of extending from Token-level Reward to Step/Trajectory-Level
### ORPO relevance
- Our theory incorporates ORPO methods into our implicit reward framework
## Clearly Stated Future Work Concerns
- Extending experiments to other domains
- Exploring multi-objective methods
## Unresolved Concerns and Our Responses
### Small learning rate range definition
- A small learning rate is a search direction, similar to how the optimal coefficient range for KL terms still requires search
### Experimental completeness
- We detail previous work, proving consistency in model selection and exp design
- Additionally, we supplement experimental results on Qwen2.5, which also align with our theoretical conclusions
## Supplementary Results
We acknowledge ongoing concerns about Qwen2.5-3B in Table 3 and provide further results here. It improved as expected.
|  | AE 2 | | AH | MTbench | |
|:--|:--|:--|:--|:--|:--|
| | LC-win | Win | Win | GPT-4 | GPT-4-turbo |
| SFT | 5.7 | 3.32 | 3.7 | 5.9 | 5.8 |
| DPO | 10.79 | 8.73 | 11.6 | 7 | 6.9 |
| P-SFT | 6.45 | 3.72 | 3.2 | 6.2 | 6.1 |
| P-DPO | 11.28 | 9.03 | 14.3 | 7 | 6.8 |
| SH-SFT | 4.7 | 3.2 | 4.5 | 6.3 | 6.1 |
| SH-DPO | 11.08 | 9.07 | 13.5 | 6.9 | 6.6 |

---

### Decision · Program_Chairs · 2025-09-17

**Decision:**

Accept (poster)

**Comment:**

The authors proposed a unified framework to bridge SFT and DPO, the two training paradigms in current foundation model training. Reviewers agree the theoretical contribution is crucial to the community. There are some argumentative discussion during rebuttal, we ask the authors to extend the experiments as required by different reviewers.